_Article_

EMBO
Molecular Medicine

# Targeting cIAP2 in a novel senolytic strategy prevents glioblastoma recurrence after radiotherapy

Nozomi Tomimatsu[1,11], Luis Fernando Macedo Di Cristofaro [1,11], Suman Kanji[1,11], Lorena Samentar[1,11],
Benjamin Russell Jordan [1,2], Ralf Kittler [3], Amyn A Habib[4], Jair Machado Espindola-Netto[5],
Tamara Tchkonia [5], James L Kirkland[6], Terry C Burns[7], Jann N Sarkaria[8], Andrea Gilbert[9], John R Floyd[1],
Robert Hromas[10], Weixing Zhao [2], Daohong Zhou [2], Patrick Sung[2], Bipasha Mukherjee [1✉] &
Sandeep Burma [1,2✉]

## Abstract

Glioblastomas (GBM) are routinely treated with high doses of ionizing radiation (IR), yet these tumors recur quickly, and the recurrent tumors are highly therapy resistant. Here, we report that IR-induced senescence of tumor cells counterintuitively spurs GBM recurrence, driven by the senescence-associated secretory phenotype (SASP). We find that irradiated GBM cell lines and patient derived xenograft (PDX) cultures senesce rapidly in a p21-dependent manner. Senescent glioma cells upregulate SASP genes and secrete a panoply of SASP factors, prominently interleukin IL-6, an activator of the JAK-STAT3 pathway. These SASP factors collectively activate the JAK-STAT3 and NF-κB pathways in non-senescent GBM cells, thereby promoting tumor cell proliferation and SASP spreading. Transcriptomic analyses of irradiated GBM cells and the TCGA database reveal that the cellular inhibitor of apoptosis protein 2 (cIAP2), encoded by the _BIRC3_ gene, is a potential survival factor for senescent glioma cells. Senescent GBM cells not only upregulate _BIRC3_ but also induce _BIRC3_ expression and promote radioresistance in non-senescent tumor cells. We find that second mitochondria-derived activator of caspases (SMAC) mimetics targeting cIAP2 act as novel senolytics that trigger apoptosis of senescent GBM cells with minimal toxicity towards normal brain cells. Finally, using both PDX and immunocompetent mouse models of GBM, we show that the SMAC mimetic birinapant, administered as an adjuvant after radiotherapy, can eliminate senescent GBM cells and prevent the emergence of recurrent tumors. Taken together, our results clearly indicate that significant improvement in GBM patient survival may become possible in the clinic by eliminating senescent cells arising after radiotherapy.

**Keywords** Glioblastoma Recurrence; Radiation Therapy; Therapy-induced Senescence; Senescence-associated Secretory Phenotype; Senolytic

Therapy
**Subject Categories** Cancer; Cell Cycle

## Introduction

Glioblastomas (GBMs) are aggressive and lethal brain cancers originating from neural stem cell (NSCs), oligodendrocyte precursor cells (OPCs) and NSC-derived astrocytes (Furnari et al, 2007; Yao et al, 2018). GBM patients are routinely treated with up to 60 Gy of ionizing radiation (IR) (Weller et al, 2014) and concurrent and adjuvant chemotherapy with temozolomide (TMZ) (Stupp et al, 2009). Unfortunately, GBMs almost always recur after a relatively short period of time following radiotherapy, with median progression-free survival ranging from 5.5 to 13 months (Hegi et al, 2005), and the recurrent tumor is usually more aggressive (Barbagallo Jenkinson and Brodbelt, 2008) and therapy resistant (Osuka and Van Meir, 2017). Therefore, a mechanistic understanding of the basis of GBM recurrence after radiotherapy is essential in order to improve patient survival.

IR exerts tumor control by triggering cell death or senescence of tumor cells. Apart from senescence of tumor cells, IR can also induce senescence of non-neoplastic cells in the tumor microenvironment (Prasanna et al, 2021). Senescence is a state of permanent cell cycle arrest that is enforced by p53-p21 or p16Ink4a-Rb tumor suppressor pathways in response to telomere shortening, oncogenic activation, or DNA damage (Hernandez-Segura Nehme and Demaria, 2018). Though senescence is a well-established anticancer barrier, recent research has revealed a paradoxical pro-tumor role of senescence. This is because senescent cells secrete

[1]Department of Neurosurgery, University of Texas Health, San Antonio, TX, USA. [2]Department of Biochemistry and Structural Biology, University of Texas Health, San Antonio, TX, USA. [3]Eugene McDermott Center for Human Growth and Development, University of Texas Southwestern Medical Center, Dallas, TX, USA. [4]Department of Neurology, University of Texas Southwestern Medical Center, Dallas, TX, USA. [5]Department of Physiology and Biomedical Engineering, Mayo Clinic, Rochester, MN, USA. [6]Department of Medicine, Mayo Clinic, Rochester, MN, USA. [7]Department of Neurological Surgery, Mayo Clinic, Rochester, MN, USA. [8]Department of Radiation Oncology, Mayo Clinic, Rochester, MN, USA. [9]Department of Pathology, University of Texas Health, San Antonio, TX, USA. [10]Department of Medicine, University of Texas Health, San Antonio, TX, USA. [11]These authors contributed equally: Nozomi Tomimatsu, Luis Fernando Macedo Di Cristofaro, Suman Kanji, Lorena Samentar. ✉E-mail: mukherjeeb@uthscsa.edu; burma@uthscsa.edu

cytokines, chemokines, growth factors, extracellular matrix components and matrix-modifying metalloproteases, collectively referred to as the senescence-associated secretory phenotype (SASP), which promote inflammation and tumorigenesis (Campisi, 2013; Coppe et al, 2010; Schmitt Wang and Demaria, 2022). Importantly, senescent stromal cells have been shown to promote the growth of neighboring tumor cells via SASP (Bavik et al, 2006; Coppe et al, 2006; Coppe et al, 2008; Demaria et al, 2017; Liu and Hornsby, 2007; Parrinello et al, 2005; Tsai et al, 2005). Moreover, senescent tumor cells could contribute to tumor growth and relapse via paracrine cooperation with neighboring non-senescent, proliferating tumor cells (Prasanna et al, 2021). Hence, there is burgeoning interest in the development of "senolytic" compounds for cancer therapy that would specifically eliminate senescent cells and attenuate SASP while sparing healthy cells (Chaib Tchkonia and Kirkland, 2022; Paez-Ribes et al, 2019; Wang Lankhorst and Bernards, 2022). It has been proposed that such senolytics should ideally be deployed after chemotherapy and radiotherapy in a "one-two punch" approach wherein therapy-induced senescence (TIS) of tumor cells renders them vulnerable to subsequent senolytic treatment (Prasanna et al, 2021; Riviere-Cazaux et al, 2023a).

During GBM therapy, a 2–3 cm margin of normal brain tissue is irradiated along with the tumor in order to eliminate infiltrating tumor cells (Wernicke et al, 2016). However, a majority of recurrences occur at or near the margin, indicating that some of the residual tumor cells lurking therein are able to re-proliferate in spite of the high doses of radiation used. IR could trigger senescence of both non-neoplastic and neoplastic cells within the radiation volume, with both types of senescent cells potentially spurring recurrence via SASP. Indeed, we reported previously that senescent astrocytes generated by radiation exposure promote the growth and invasiveness of adjacent malignant cells in mouse models of GBM (Fletcher-Sananikone et al, 2021). Specifically, we found that tumor growth and invasiveness were spurred by senescent astrocyte-derived HGF that activated the MET receptor tyrosine kinase in glioma cells, and these effects could be blunted by the clearance of senescent astrocytes using senolytic approaches. These findings indicated that the high doses of radiation used for GBM therapy could counterintuitively contribute to tumor recurrence via induction of senescence in non-neoplastic brain cells. In this study, we asked if radiotherapy could similarly drive brain tumor cells into a senescent state, whether senescent tumor cells might spur the proliferation or therapy resistance of their non-senescent counterparts via SASP, and whether these senescent cells harbor vulnerabilities that could be exploited to prevent GBM recurrence.

## Results

### Irradiated GBM cells undergo senescence via induction of p21, upregulate SASP genes, and secrete SASP factors

As senescence is induced by the p16$^{INK4a}$-pRB and/or p53-p21 tumor suppressor pathways (Campisi, 2013), we first evaluated if IR can induce senescence in GBM cells that are frequently mutated in components of these pathways. The CDKN2A/CDKN2B locus, which codes for p16, is homozygously deleted in more than 60% of GBM (Brennan et al, 2013). On the other hand, p21 is rarely

mutated in cancers including GBM (Abbas and Dutta, 2009), though mutations in p53 are seen in about 25% of GBM (Brennan et al, 2013). We therefore initially focused on four human GBM cell lines differing in p53 status: LN229 (heterozygous mutation), A172 (heterozygous SNP), U118 (homozygous mutation) and U87 (wild type) (Patyka et al, 2016). All four lines, as with 90% of GBM cells in culture (Furnari et al, 2007), are nullizygous for p16. Cells were exposed to 10 Gy of X-rays and evaluated after 10 days for induction of senescence using multiple markers recommended by the International Cell Senescence Association (ICSA)—senescence-associated-beta-galactosidase (SA-β-gal) staining, loss of nuclear Lamin B1, loss of the proliferation marker Ki67, and upregulation of p21 (Gorgoulis et al, 2019). We found that cells surviving after irradiation exhibited a high degree of SA-β-gal-positivity along with loss of Lamin B1- and Ki67-staining (Figs. 1A and EV1A,B). In addition, all four lines, regardless of p53 status, showed high levels of p21 expression relative to mock-irradiated cells (Fig. 1B).

Next, we carried out RNA-seq analysis of mock-irradiated or irradiated (senescent) LN229 and A172 cells. The RNA-seq data revealed a very striking similarity in the gene expression patterns of LN229 and A172 cells, when comparing the 100 most differentially expressed genes (DEGs) (Fig. EV1C). We used the RNA-seq data to quantify the senescence score of these cells, on a scale of 0 (non-senescent) to 1 (truly senescent) by using the Senescence Classifier from Cancer SENESCopedia (Jochems et al, 2021). While mock-irradiated cells had a score of 0, irradiated LN229 and A172 cells had scores of 0.91 and 0.99, respectively, thereby re-confirming that the irradiated cells were truly senescent (Fig. EV1D). Gene Set Enrichment Analysis (GSEA) revealed common enriched pathways in senescent LN229 and A172 cells, including "SASP", "cytokine-cytokine receptor interaction", "JAK-STAT signaling", and "NF-κB signaling" (Fig. EV1E). We evaluated the expression of a set of core SASP genes that have been commonly shown to be induced in senescent cells (Coppe et al, 2010; Suryadevara et al, 2024), and found these to be upregulated in senescent LN229 and A172 cells (Fig. 1C). Furthermore, we confirmed the induction of these 10 genes, including those coding for the major SASP components IL-6 and IL-8, in all four senescent GBM cell lines by qRT-PCR (Fig. 1D).

To confirm that the senescent cells are capable of secreting the cytokines and chemokines that constitute the SASP, we collected conditioned media (CM) from mock-irradiated and irradiated GBM cells and measured the levels of a panel of 41 secreted proteins that define the SASP using a magnetic bead-based immunoassay based on Luminex FlexMap3D multiplexing platform (Xu et al, 2015). We found that most of the SASP proteins were significantly more abundant in CM from senescent GBM cells than from naive GBM cells (Appendix Table S1). A common subset of 24 SASP proteins was found to be upregulated in all four lines, which included both IL-6 and IL-8 (Fig. 1E). By utilizing the TRRUST v2 database (Han et al, 2018), we found that signaling networks triggered by these 24 SASP factors could potentially lead to the activation of a number of transcription factors with the top five hits being STAT3, NF-κB, FOS/JUN, SP1 and ETS2 (Fig. EV1F). These results indicate that GBM cells, despite being p16-null, can undergo bona fide senescence which is likely initiated and maintained by the induction of p21 via p53-independent mechanisms (Abbas and Dutta, 2009; Georgakilas Martin and Bonner, 2017), and that these senescent cells secrete SASP factors that generate a pro-tumorigenic milieu.

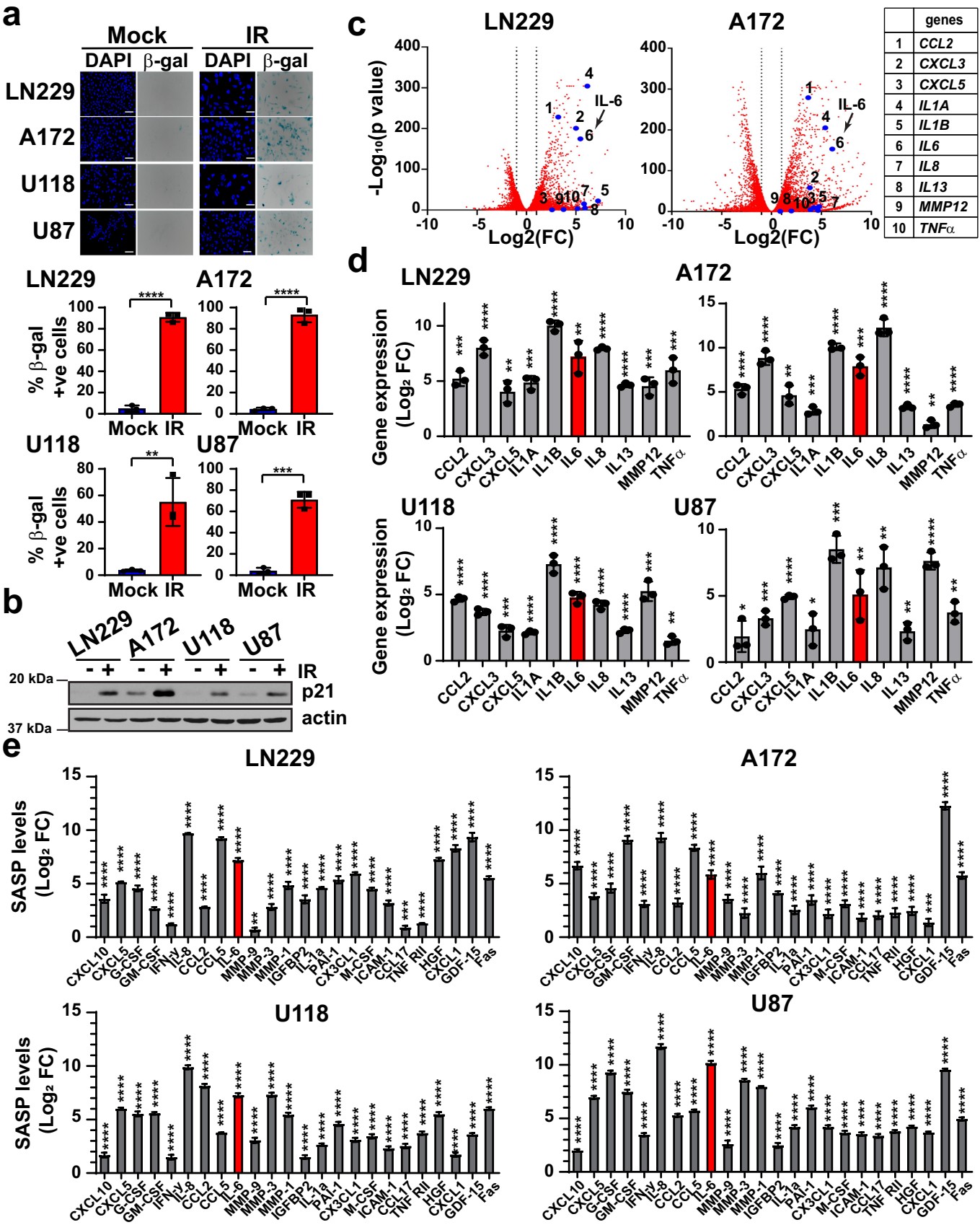

**Figure 1. Irradiated GBM cells undergo senescence, upregulate SASP genes, and secrete SASP factors.**

(A) Representative images of SA-β-gal staining of GBM cell lines mock-irradiated (Mock) or irradiated (IR) with 10 Gy of X-rays and then allowed to recover for 10 days ($n = 3$ with at least 100 nuclei scored for each replicate). Nuclei are stained with DAPI (blue). Plots show mean percentage of SA-β-gal-positive cells $+/-$ SD. A two-tailed Student's $t$ test was performed; LN229 $P = 0.000009$, A172 $P = 0.000027$, U118 $P = 0.000141$, U87 $P = 0.007801$. Scale bar, 100 µm. (B) Whole cell extracts from mock-irradiated or irradiated GBM cell lines were western blotted with anti-p21 antibody. Actin serves as loading control. (C) Volcano plot representation of differentially expressed genes from mock-irradiated $vs.$ irradiated LN229 or A172 cells ($n = 3$), as assessed by RNA sequencing 10 days after treatment with 10 Gy of X-rays. Differential gene expression analysis was performed using edgeR. Gene-specific dispersions were estimated using a tagwise dispersion model and statistical significance was assessed using the likelihood ratio test (LRT). The false discovery rate (FDR) was calculated using the Benjamini–Hochberg procedure. Genes with an FDR < 0.05 with $\log_2$ fold change ($\text{Log}_2$ FC) cutoff of $-1$ and 1 were considered differentially expressed, as denoted by dashed lines. Core SASP genes that were further validated by qRT-PCR are represented by numbers. (D) Total RNA was isolated from GBM cells 10 days after irradiation with 10 Gy of X-rays or from mock-irradiated cells and expression of SASP genes assessed by qRT-PCR ($n = 3$ biological replicates comprising three technical replicates each). Plots show mean fold change ($\text{Log}_2$ FC) in gene expression $+/-$ SD of SASP-related genes in irradiated GBM cells relative to mock-irradiated cells. A two-tailed Student's $t$ test was performed; $*P < 0.05$; $**P < 0.01$; $***P < 0.001$; $****P < 0.0001$ (please refer to Appendix Table S4 for the exact p values). (E) Conditioned media were collected from GBM cells 10 days after irradiation with 10 Gy of X-rays ($n = 3$ biological replicates comprising three technical replicates each), and cytokine levels were measured by multiplex bead-based immunoassay. Plots show mean fold change ($\text{Log}_2$ FC) in cytokine levels $+/-$ SD in media from irradiated cells relative to mock-irradiated cells. A two-tailed Student's $t$ test was performed; $**P < 0.01$; $***P < 0.001$; $****P < 0.0001$ (please refer to Appendix Table S4 for the exact $P$ values). Source data are available online for this figure.

## Senescent GBM cells activate the JAK-STAT3 and NF-κB pathways in naive GBM cells to promote tumor cell proliferation and SASP spreading

Of the prototypical SASP factors that we had initially queried (Fig. 1C), IL-6 was of particular interest to us as this cytokine has prominent tumor-promoting roles via activation of the JAK-STAT3 pathway (Johnson O'Keefe and Grandis, 2018). We carried out media transfer experiments where we added CM from non-senescent (CM-NS) or senescent (CM-SEN) GBM cells to naive GBM cells. We found that CM-SEN could activate the JAK-STAT3 pathway in recipient cells as evaluated by phosphorylation of STAT3 at tyrosine 705 (Fig. 2A), validating predictions from TRRUST analysis of SASP factors secreted by senescent GBM cells (Fig. EV1F). Phosphorylation of STAT3 could be blocked by neutralizing IL-6 in CM-SEN with an anti-IL-6 antibody or by treating the naive GBM cells with the JAK inhibitor ruxolitinib (Johnson O'Keefe and Grandis, 2018) (Fig. 2B), consistent with our finding that ruxolitinib inhibits the SASP in other senescent cell types and alleviates frailty in old mice (Xu et al, 2015). Importantly, we found that CM-SEN could stimulate the proliferation of naive GBM cells, as measured by BrdU pulsing, and this was reversed by treatment of the naive cells with ruxolitinib (Fig. 2C). In addition, we found that CM-SEN could also activate the NF-κB pathway in recipient GBM cells, seen as hyperphosphorylation of p65/RELA at serine 536 (Hayden and Ghosh, 2012) (Fig. EV2A), validating predictions from TRRUST analysis of proteins secreted by senescent GBM cells (Fig. EV1F). As NF-κB induces the transcription of a majority of SASP genes (Ohanna et al, 2011), this observation implied that CM-SEN could induce the SASP in naive GBM cells. Indeed, upon treating naive GBM cells with CM-SEN, we saw upregulation of a number of SASP genes, including *IL6* and *IL8* (Fig. EV2B). Taken together, these findings suggest that senescent GBM cells can stimulate the proliferation of their non-senescent counterparts via activation of the JAK-STAT3 pathway, and that SASP can spread from senescent to non-senescent GBM cells, thereby amplifying the pro-tumorigenic effects of radiation-induced senescence.

## Senescent GBM cells upregulate the anti-apoptotic gene *BIRC3* as a survival mechanism and promote *BIRC3* gene expression in naive GBM cells

Activation of anti-apoptotic pathways is not only a hallmark of cellular senescence (Gorgoulis et al, 2019) but is also a vulnerability that could

be targeted for therapy (Schmitt Wang and Demaria, 2022; Wang Lankhorst and Bernards, 2022; Zhu et al, 2015). With this concept in mind, we next assessed expression levels of BCL-2 and inhibitor of apoptosis (IAP) family members (Carneiro and El-Deiry, 2020; Fulda and Vucic, 2012; Singh Letai and Sarosiek, 2019) in the RNA-seq datasets from naive vs. senescent LN229 and A172 cells. We found two genes—*BCL2L10* and *BIRC3* (which codes for cIAP2)—with $\log_2$ fold change in gene expression greater than 3 in both lines. *BCL2L10* showed 5.59 and 3.93 $\log_2$ fold change while *BIRC3* showed 5.76 and 3.49 $\log_2$ fold change in transcript levels in LN229 and A172 cells, respectively (Fig. 3A). Of these two genes, we chose to focus on *BIRC3* for the following reasons: (a) Upon plotting the average abundance of transcripts in counts per million (CPM) against fold change in gene expression (McDermaid et al, 2019), it became apparent that *BIRC3* transcripts were significantly more abundant than *BCL2L10* transcripts in both GBM cell lines (Fig. 3B); (b) In both The Cancer Genome Atlas (TCGA) and Chinese Glioma Genome Atlas (CGGA) (Zhao et al, 2021) databases of glioma patients, high *BIRC3* transcript levels correlated with poor prognosis (Fig. 3C) while no such correlation was seen with *BCL2L10* levels (Fig. EV3A); (c) cIAP2 is a highly druggable target that can be inhibited by SMAC mimetics (Bai Smith and Wang, 2014) while BCL2L10 is not druggable to the best of our knowledge.

To validate the upregulation of *BIRC3* gene expression indicated by the RNA-seq data, we irradiated GBM cells and evaluated them after 10 days by qRT-PCR and western blotting. We found several-fold increases in *BIRC3* transcript levels in all four irradiated cell lines along with corresponding increases in cIAP2 protein levels (Fig. 3D,E), in concordance with the RNA-seq data (Fig. 3B). In contrast, we saw no or low levels of induction of the related *BIRC2* gene (Fig. EV3B) and the cIAP1 protein encoded by this gene (Fig. EV3C). Transcript and protein levels of cIAP2 in senescent GBM cells could be reduced by treatment with the IκB kinase (IKK) inhibitor BMS-345541 (Gilmore and Herscovitch, 2006; Waelchli et al, 2006) (Figs. 3F,G and EV3D,E), in keeping with previous reports showing that *BIRC3* gene transcription is dependent on NF-κB (Karin and Lin, 2002). Interestingly, we found that CM-SEN could increase *BIRC3* transcript and cIAP2 protein levels in naive GBM cells (Figs. 3H,I and EV3F,G). The induction of cIAP2 in naive GBM cells was likely due to the activation of the NF-κB pathway by SASP factors in CM-SEN (Fig. EV2A), as cIAP2 induction could be blocked by treatment of the naive cells with

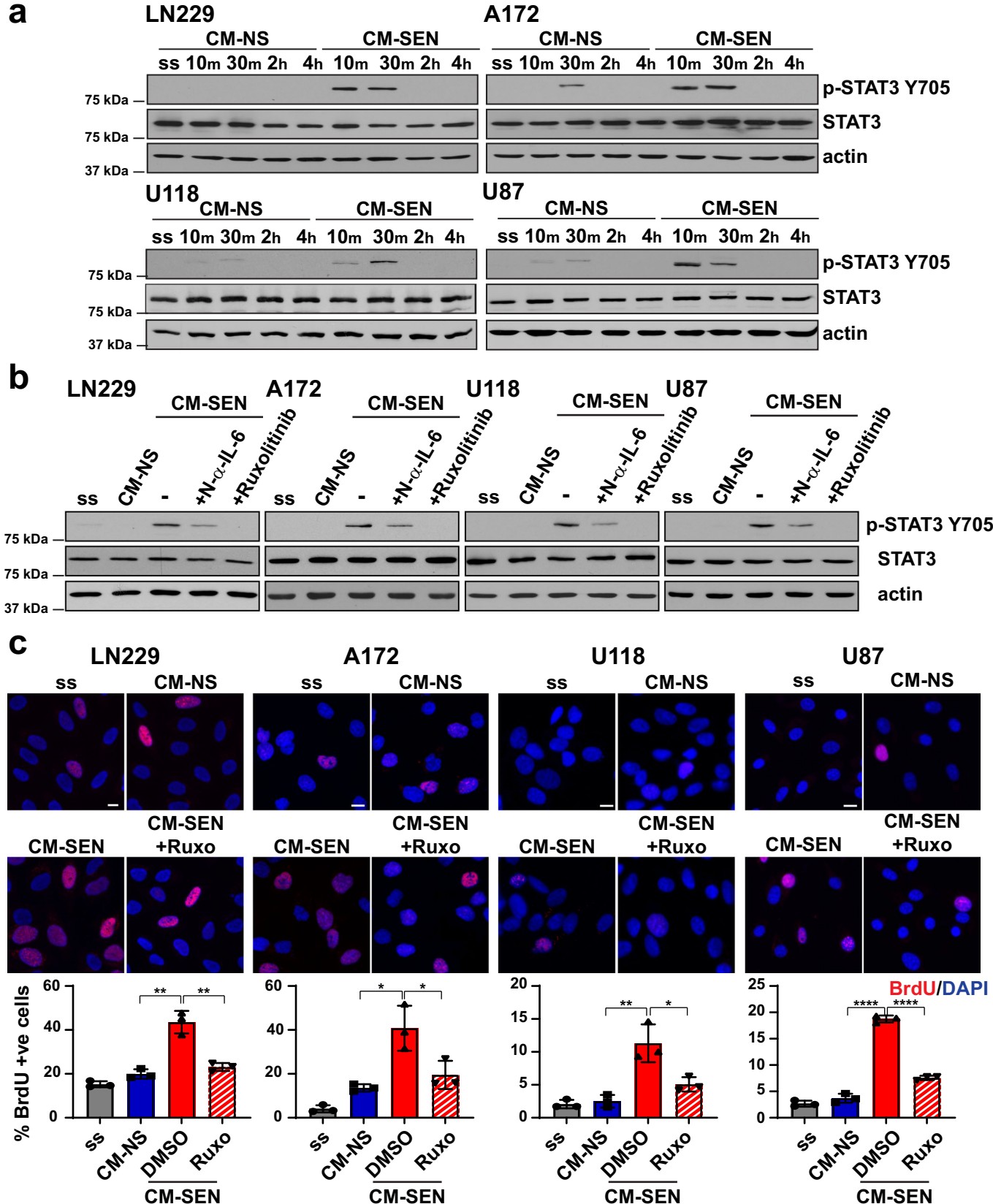

◀

**Figure 2. Senescent GBM cells activate the JAK-STAT3 pathway in naive GBM cells.**

(A) Naive GBM cells were exposed to conditioned media from senescent (CM-SEN) or non-senescent (CM-NS) GBM cells for the indicated times, and activation of the JAK-STAT3 pathway assessed by western blotting with anti-phopho-STAT3 (Y705) antibody. Actin serves as loading control. Recipient cells were serum starved (ss) before addition of CM. (B) Phosphorylation of STAT3 after exposure of naive cells to CM-SEN for 30 min was assessed by western blotting after either neutralization of IL-6 in the conditioned media with an anti-IL-6 antibody (N-α-IL-6) or treatment of recipient GBM cells with the JAK inhibitor ruxolitinib, as indicated. (C) Serum starved (ss) GBM cells were pulsed with BrdU after exposure to CM-NS, CM-SEN, or CM-SEN in the presence of ruxolitinib (CM-SEN+Ruxo), and immunofluorescence stained with anti-BrdU antibody (red), as indicated ($n = 3$ with at least 100 nuclei scored for each replicate). Nuclei are stained with DAPI (blue). Plots show mean percentages of BrdU-positive cells $+/-$ SD. A two-tailed Student's $t$ test was performed; LN229 $P = 0.00183, 0.0028$; A172 $P = 0.01061, 0.03826$; U118 $P = 0.00736, 0.02480$; U87 $P = 0.00002, 0.00002$, respectively. Scale bar, 10 μm. Source data are available online for this figure.

BMS-345541 (Figs. 3I,J and EV3G,H). We speculate that senescent GBM cells may not only rely on cIAP2 for their survival but might also augment survival of their non-senescent counterparts via SASP-mediated cIAP2 induction. In support of this concept, we found that CM-SEN could augment the radioresistance of naive GBM cells in colony survival assays (Fig. EV3I). Taken together, these data indicate that senescent GBM cells upregulate *BIRC3* to promote their own survival and also that of neighboring tumor cells, raising the possibility of targeting cIAP2 in a novel senolytic approach to eliminate senescence and the SASP.

## SMAC mimetics can eliminate senescent GBM cells by targeting cIAP2

While targeting BCL-2 family members has been a common strategy for eliminating senescent cells (Schmitt Wang and Demaria, 2022; Wang Lankhorst and Bernards, 2022), our results indicate that cIAP2 could be a novel candidate for senolytic therapy. SMAC mimetic compounds (SMCs) bind to cIAP1 and cIAP2, thereby triggering auto-ubiquitination and degradation of these anti-apoptotic proteins (Dueber et al, 2011). To test if SMCs can selectively eliminate senescent cells, we irradiated GBM cells and treated them 10 days later with birinapant or LCL161, both drugs being SMCs that are currently in clinical trials (Morrish Brumatti and Silke, 2020). We found that both drugs could markedly reduce cIAP2 levels within 30 min, though cIAP2 increased to pre-treatment levels by 24 h (Figs. 4A and EV4A). Consequently, both drugs triggered apoptosis in senescent GBM cells by 24 h, as evidenced by cleavage of caspase-8 (Figs. 4A and EV4A). Induction of cell death by birinapant was confirmed by propidium iodide (PI) exclusion (Crowley et al, 2016) and LIVE/DEAD™ assays (Appendix Fig. S1a,b). Next, we treated naive or senescent (day 10 post-IR) GBM cells with SMCs (replacing media with the drug every third day) and evaluated cell survival over a 9-day period by crystal violet staining and the MTT assay. While naive cells were not affected by either drug, senescent GBM cells were progressively eliminated by intermittent drug treatment (Figs. 4B,C and EV4B,C). Though SMCs can target both cIAP1 and cIAP2 (Dueber et al, 2011), cell death was likely due to degradation of cIAP2, as we did not observe substantial upregulation of cIAP1 in senescent GBM cells (Fig. EV3C). Moreover, siRNA-mediated knockdown of cIAP2 resulted in cell death (Appendix Fig. S1c–e), confirming that cIAP2 is vulnerability in senescent GBM cells that can be targeted by SMCs. In sum, these results show that SMAC mimetics can act as novel senolytics as they specifically eliminate senescent GBM cells arising after radiotherapy without affecting naive, unirradiated cells.

## Targeting cIAP2 ablates radiation-induced senescence in GBM patient derived xenograft (PDX) cultures

Next, we sought to validate our findings of radiation-induced senescence and SASP, as well as the paracrine effects of senescent cells, in short term cultures from GBM PDXs that maintain key molecular and histopathological features of their parental tumors (Carlson et al, 2011). We chose six lines differing in p53 and p16 status from the Mayo panel of genetically characterized GBM PDXs (Appendix Table S2) (Vaubel et al, 2020). All six lines underwent senescence within 10 days of irradiation, as evidenced by increased SA-β-gal staining, upregulation of a panel of SASP genes, and induction of p21 or p16 (Fig. 5A–C). The one line that did not show p21 induction (GBM123) is p16-positive and showed upregulation of p16 instead (Fig. 5B). In media transfer experiments, we found that CM-SEN from senescent PDX cultures could activate the JAK-STAT3 pathway in the corresponding naive cultures (Fig. 5D), validating results obtained with the GBM cell lines (Fig. 2A), and this could be blocked by treating the naive cultures with ruxolitinib (Fig. EV5A). Accordingly, we found that CM-SEN could stimulate the proliferation of naive PDX cultures, and this could be blunted by treatment of naive cells with ruxolitinib (Fig. EV5B; Appendix Fig. S2).

Importantly, we observed increases in *BIRC3* transcript and cIAP2 protein levels in all six PDX lines after induction of senescence (Fig. 6A–C). In contrast, most PDX cultures showed low or insignificant increases in *BIRC2* transcript and cIAP1 protein levels (Appendix Fig. S3a,b). Interestingly, CM-SEN from senescent PDX cultures could induce cIAP2 in the corresponding naive cultures (Fig. 6D), validating results obtained with the GBM cell lines (Figs. 3H and EV3F). Treatment of senescent PDX cultures with birinapant led to rapid loss of cIAP2 (Fig. 6A,B). Accordingly, senescent GBM cells could be killed by birinapant treatment, while naive GBM cells were unaffected by the drug, as assessed by crystal violet staining as well as by the MTT assay (Figs. 6E and EV5C). Primary cultures of normal human astrocytes (NHA) showed undetectable levels of cIAP2 by western blotting (Appendix Fig. 4a) and were accordingly completely resistant to birinapant treatment (Appendix Fig. S4b). These results, in aggregate, clearly demonstrate that diverse GBM PDX cultures, irrespective of their genetic signatures, undergo senescence and exhibit upregulation of SASP genes after radiation exposure, potentially reflecting the responses of cells within irradiated human tumors. Importantly, these results indicate that the induction of cIAP2 in senescent GBM cells is a common vulnerability that could be targeted with SMAC mimetics without affecting normal CNS cells such as astrocytes.

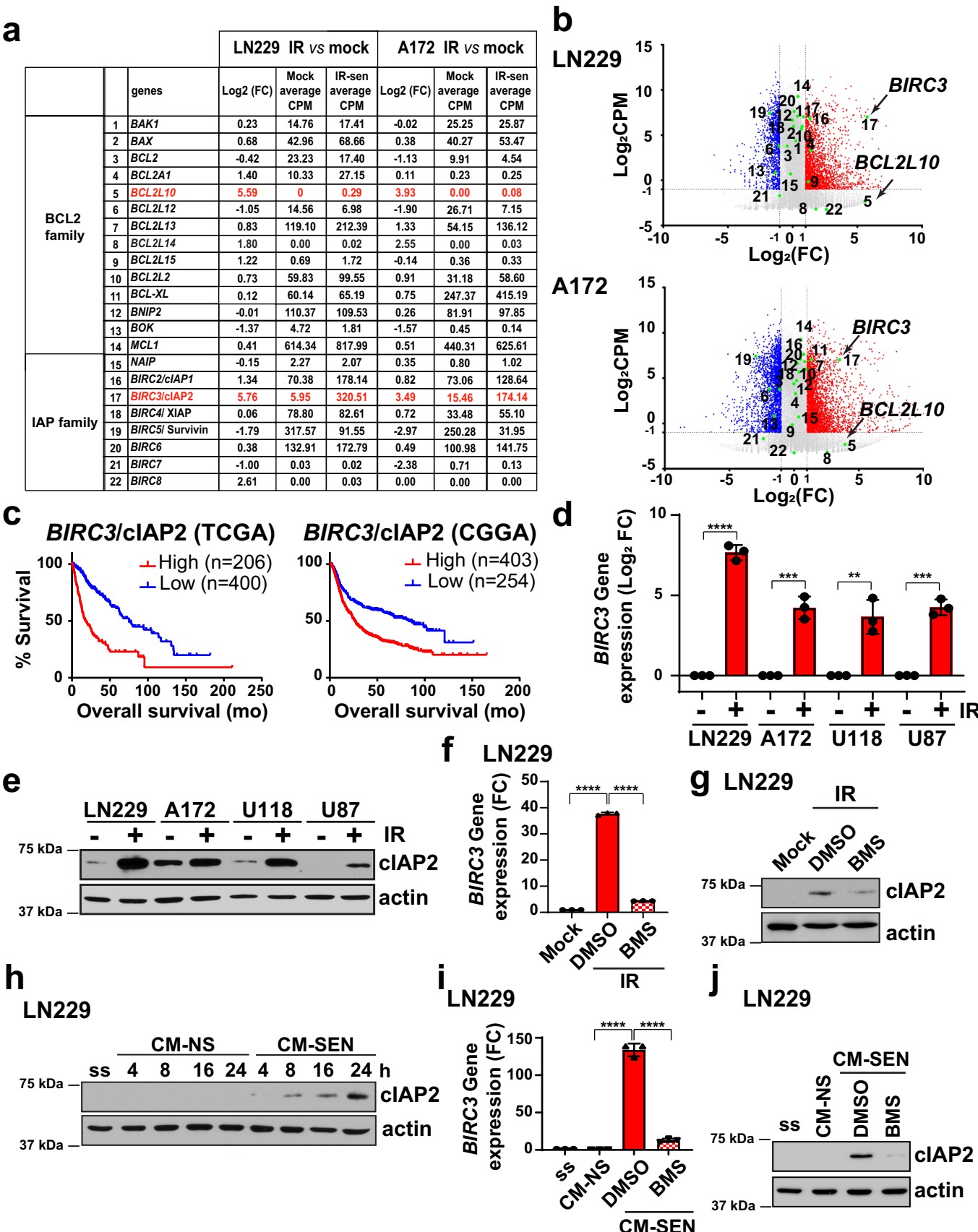

**a**

|  |  | LN229 IR vs mock | | | A172 IR vs mock | | |
|---|---|---|---|---|---|---|---|
|  | genes | Log2 (FC) | Mock average CPM | IR-sen average CPM | Log2 (FC) | Mock average CPM | IR-sen average CPM |
| BCL2 family | 1 | BAK1 | 0.23 | 14.76 | 17.41 | -0.02 | 25.25 | 25.87 |
|  | 2 | BAX | 0.68 | 42.96 | 68.66 | 0.38 | 40.27 | 53.47 |
|  | 3 | BCL2 | -0.42 | 23.23 | 17.40 | -1.13 | 9.91 | 4.54 |
|  | 4 | BCL2A1 | 1.40 | 10.33 | 27.15 | 0.11 | 0.23 | 0.25 |
|  | 5 | BCL2L10 | 5.59 | 0 | 0.29 | 3.93 | 0.00 | 0.08 |
|  | 6 | BCL2L12 | -1.05 | 14.56 | 6.98 | -1.90 | 26.71 | 7.15 |
|  | 7 | BCL2L13 | 0.83 | 119.10 | 212.39 | 1.33 | 54.15 | 136.12 |
|  | 8 | BCL2L14 | 1.80 | 0.00 | 0.02 | 2.55 | 0.00 | 0.03 |
|  | 9 | BCL2L15 | 1.22 | 0.69 | 1.72 | -0.14 | 0.36 | 0.33 |
|  | 10 | BCL2L2 | 0.73 | 59.83 | 99.55 | 0.91 | 31.18 | 58.60 |
|  | 11 | BCL-XL | 0.12 | 60.14 | 65.19 | 0.75 | 247.37 | 415.19 |
|  | 12 | BNIP2 | -0.01 | 110.37 | 109.53 | 0.26 | 81.91 | 97.85 |
|  | 13 | BOK | -1.37 | 4.72 | 1.81 | -1.57 | 0.45 | 0.14 |
|  | 14 | MCL1 | 0.41 | 614.34 | 817.99 | 0.51 | 440.31 | 625.61 |
| IAP family | 15 | NAIP | -0.15 | 2.27 | 2.07 | 0.35 | 0.80 | 1.02 |
|  | 16 | BIRC2/cIAP1 | 1.34 | 70.38 | 178.14 | 0.82 | 73.06 | 128.64 |
|  | 17 | BIRC3/cIAP2 | 5.76 | 5.95 | 320.51 | 3.49 | 15.46 | 174.14 |
|  | 18 | BIRC4/ XIAP | 0.06 | 78.80 | 82.61 | 0.72 | 33.48 | 55.10 |
|  | 19 | BIRC5/ Survivin | -1.79 | 317.57 | 91.55 | -2.97 | 250.28 | 31.95 |
|  | 20 | BIRC6 | 0.38 | 132.91 | 172.79 | 0.49 | 100.98 | 141.75 |
|  | 21 | BIRC7 | -1.00 | 0.03 | 0.02 | -2.38 | 0.71 | 0.13 |
|  | 22 | BIRC8 | 2.61 | 0.00 | 0.03 | 0.00 | 0.00 | 0.00 |

**b** LN229

A172

**c** BIRC3/cIAP2 (TCGA)  — High (n=206)  — Low (n=400)

BIRC3/cIAP2 (CGGA)  — High (n=403)  — Low (n=254)

**d**

**e**

**f** LN229

**g** LN229

**h** LN229

**i** LN229

**j** LN229

◀ **Figure 3. Senescent GBM cells upregulate the anti-apoptotic gene *BIRC3* and induce *BIRC3* in naive cells.**

(A) List of BCL-2 and IAP family members showing fold changes (Log$_2$ FC) in gene expression in irradiated (IR) LN229 or A172 cells relative to mock-irradiated (mock) cells along with mRNA transcripts levels in counts per million (CPM), as assessed by RNA sequencing 10 days after irradiation with 10 Gy of X-rays ($n = 3$). Genes with Log$_2$ fold change greater than 3 are highlighted in red. (B) MA plot showing abundance of transcripts represented by Log$_2$ CPM ($y$ axis) *vs.* Log$_2$ fold change in gene expression ($x$ axis) in irradiated cells relative to mock-irradiated cells ($n = 3$ biological replicates per condition for each cell line). BCL-2 and IAP family gene members are denoted with numbers. Genes with log$_2$ fold change (Log$_2$ FC) cutoff of $-1$ and 1, and Log$_2$ CPM higher than $-1$ were considered differentially expressed, as denoted by dashed lines. (C) Kaplan–Meier curve showing correlation of higher *BIRC3* expression levels with poor prognosis in GBMLGG patients in TCGA ($n = 606$) and CGGA ($n = 657$) cohorts as evidenced by Hazard Ratio (logrank) of 3.441 and 1.811, respectively. $P < 0.0001$ for both plots. (D) Plot shows mean relative expression of *BIRC3* $+ / - $ SD in mock-irradiated vs. irradiated (IR) GBM cells 10 days after exposure to 10 Gy of X-rays, as assessed by qRT-PCR ($n = 3$ biological replicates comprising three technical replicates). A two-tailed Student's $t$ test was performed; exact $P$ values from left to right: 0.00001, 0.00048, 0.00397, 0.00012. (E) Whole cell extracts from mock-irradiated or irradiated GBM cell lines were western blotted with anti-cIAP2 antibody. Actin serves as loading control. (F) Senescent LN229 cells (10 days after exposure to 10 Gy) were treated with the IKK inhibitor BMS-345541 (BMS) or DMSO as control for 72 h ($n = 3$ biological replicates comprising three technical replicates), and mean relative expression of *BIRC3* $+ / - $ SD was assessed by qRT-PCR (a two-tailed Student's $t$ test was performed; $P = 0.00000002$, 0.00000003, respectively) or (G) western blotting for cIAP2. (H) Naive LN229 cells were exposed to conditioned media from senescent (CM-SEN) or non-senescent (CM-NS) cells for the indicated times, and expression of cIAP2 assessed by western blotting. (I) Naive LN229 cells were treated with BMS-345541 or DMSO as control for 2 h before exposure to CM-SEN ($n = 3$ biological replicates comprising 3 technical replicates) and mean relative expression of *BIRC3* $+ / - $ SD was assessed by qRT-PCR (a two-tailed Student's $t$ test was performed; $P = 0.00001$, 0.00002, respectively) or (J) western blotting for cIAP2. Source data are available online for this figure.

## Birinapant is a novel senolytic that can prevent tumor recurrence in pre-clinical mouse GBM models

Encouraged by the in vitro results, we wanted to test the potential utility of birinapant as a senolytic in both PDX and immunocompetent mouse models of GBM. We chose the PDX line GBM12 for the first study as it harbors common genetic alterations seen in GBMs including p16 deletion and p53 mutation (Vaubel et al, 2020). Intracranial tumors were generated in nude mice using GBM12 cells expressing luciferase to allow monitoring of tumor growth by bioluminescent imaging (BLI), as described before (Gil del Alcazar et al, 2014). In all studies, treatment was initiated when tumors reached BLI intensities in the range of 6 to $9 \times 10^6$ photons/sec and were presumably of similar size at time of treatment. First, we mock-irradiated or cranially irradiated tumor-bearing mice with a dose of 10 Gy and harvested the tumors 5 days after IR. Upon staining of tumor cryosections, we observed a high degree of SA-β-gal-positivity in the irradiated tumors, confirming the induction of senescence in these orthotopic tumors (Fig. 7A). Next, we tested the efficacy of a 15-dose regimen of birinapant in controlling tumor growth, with treatment given every other day. The study involved four treatment arms: (i) mock treatment, (ii) birinapant alone, (iii) IR alone, and (iv) IR with adjuvant birinapant. Birinapant (30 mg/kg) or vehicle alone as control was administered by oral gavage, starting at day 5 after mock-irradiation or intracranial irradiation with a dose of 10 Gy. The dose and dosing schedule of birinapant was based upon a regimen determined to be effective in a published mouse GBM study (Beug et al, 2017) and plasma half-life (30–35 h) of birinapant in human patients in a Phase I clinical trial (Amaravadi et al, 2015). BLI radiance was recorded and plotted over time, and mice were sacrificed once they became moribund due to the tumor burden (Fig. 7B). IR alone attenuated tumor growth, albeit minimally, and the tumors recurred within 20 days post-IR (Appendix Fig. S5a). While birinapant alone showed no efficacy in controlling tumor growth, the drug significantly attenuated tumor growth rates when administered as an adjuvant after IR. Accordingly, the median survival for mock- and birinapant-treated mice was approximately 16 and 15 days, respectively (Fig. 7C). While radiation alone could increase median survival to approximately 29 days, the administration of birinapant after radiation further increased median survival to approximately 40 days, with 2 out of 9 mice surviving beyond the end of study (120 days) with no discernible

tumor recurrence. Combination treatment with radiation and birinapant was not associated with changes in body weight indicating good tolerability in animals (Appendix Fig. S5b). Next, we treated irradiated tumors with 5 cycles of adjuvant birinapant or vehicle only as control and harvested the tumors 24 h after the last treatment. We found significant induction of cIAP2 in the irradiated tumors which could be ablated by birinapant treatment, indicating that the drug exerted tumor control, at least in part, by targeting cIAP2 (Appendix Fig. S5c). Lastly, we treated irradiated tumors with 10 cycles of adjuvant birinapant or vehicle only as control and harvested the tumors 24 h after the last treatment. We found that adjuvant treatment with birinapant ablated SA-β-gal staining in the irradiated tumors, indicating that elimination of senescent cells by the drug contributed to the observed delay or prevention of recurrence (Fig. 7D).

Senescence has been reported to both suppress and promote anti-tumor immunity in a context-dependent manner (Rao and Jackson, 2016). It was, therefore, important for us to assess if elimination of senescent cells by adjuvant birinapant treatment would be overall beneficial in an immunocompetent mouse GBM model, comprising GL261 cells implanted intracranially in C57BL/6J mice (Fletcher-Sananikone et al, 2021; Oh et al, 2014). We first established that GL261 cells underwent senescence and showed increased expression of SASP genes in vitro upon irradiation (Appendix Fig. S5d,e). We also confirmed the induction of cIAP2 in senescent GL261 cells, that could be ablated by birinapant treatment (Appendix Fig. S5f,g), leading to selective killing of senescent cells in vitro (Appendix Fig. S5h,i). Next, we irradiated orthotopic GL261 tumors with a dose of 10 Gy and confirmed that these tumors underwent senescence by day 5 post-IR (Fig. 7E). We subsequently tested the effects of a 15-dose regimen of birinapant on these tumors, using the same drug and radiation doses and dosing schedules used in the GBM12 study, except that treatment was initiated when tumors reached BLI intensities in the range of 2 to $2.5 \times 10^6$ photons/sec. We found that birinapant alone showed no efficacy in controlling tumor growth (Fig. 7F; Appendix Fig. S5j) and did not improve survival of tumor-bearing mice (Fig. 7G). While radiation alone could increase median survival from 19 days to approximately 33 days, the administration of adjuvant birinapant after radiation strikingly increased median survival to ~47 days. Combination treatment with radiation and birinapant was not associated with changes in body weight indicating low

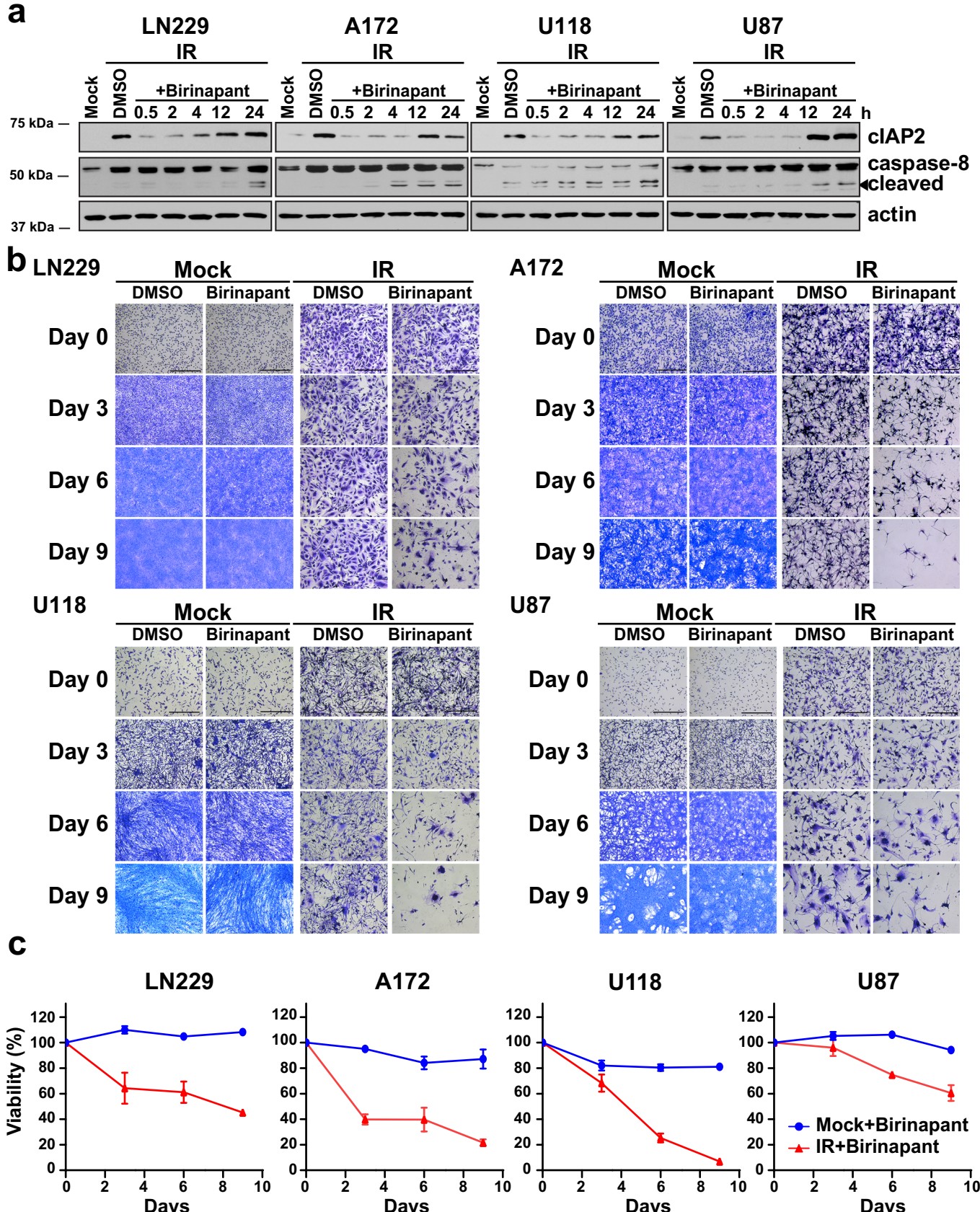

◄ **Figure 4.  The SMAC mimetic birinapant selectively eliminates senescent GBM cells.**

(**A**) GBM cells were irradiated with 10 Gy of X-rays (IR) and treated with the cIAP2 inhibitor birinapant (or DMSO as control) after 10 days for the indicated times, and cIAP2 levels and cleavage of caspase-8 were assessed by western blotting. Actin serves as loading control. (**B**) Mock-irradiated or irradiated GBM cells were treated with birinapant or DMSO as control, and the surviving cells were visualized by staining with crystal violet at the indicated times (Scale bar, 500 μm), and (**C**) viability (normalized to that of DMSO-treated cells, $n = 8$ or 9 replicates per cell line) was quantified by the MTT assay. The drug was replaced every 72 h. Plots show mean viability $+/-$ SD. Source data are available online for this figure.

toxicity in animals (Appendix Fig. S5k). As with the GBM12 tumors, we found that irradiated GL261 tumors treated with 9 cycles of adjuvant birinapant showed a reduction in SA-β-gal staining, confirming that the senolytic effects of the drug contributed to the observed thwarting of recurrence (Fig. 7H).

Taken together, our pre-clinical results imply that significant improvement in GBM therapy may become possible in the clinic with tailored senolytic therapies given after radiation, as exemplified by our novel use of SMAC mimetics to target senescent GBM cells that rely on cIAP2 for survival.

## Discussion

Although GBMs are highly radioresistant, IR still remains the most effective therapeutic modality for these tumors, as no other treatment has been efficacious except for TMZ which, unfortunately, adds only a few months to survival (Stupp et al, 2009). Thus, IR will likely remain part of the standard-of-care for GBM for the foreseeable future, and new therapeutic modalities must work in conjunction with IR. GBM recurrence after radiotherapy is a major cause of treatment failure as recurrence occurs rapidly, and the recurrent tumor is usually very resistant to further therapeutic interventions (Osuka and Van Meir, 2017). In order to develop strategies to delay recurrence or treat the recurrent tumor, it is important to understand how radiation alters both the tumor and the tumor microenvironment (Fletcher-Sananikone et al, 2021; Gupta et al, 2020; Li et al, 2015). As IR is a potent inducer of senescence (Campisi, 2013), a better understanding of how senescence influences GBM recurrence is necessary for the development of effective therapies for these deadly tumors.

Because GBMs very commonly harbor deletions of the p16 locus and frequently carry mutations in p53 (Brennan et al, 2013), we utilized a panel of p16-null GBM cell lines differing in p53 mutation status to evaluate if these cells could enter a senescent stage after irradiation. Based upon evaluation of a number of senescence markers (Gorgoulis et al, 2019), including the induction of SASP genes and secretion of SASP factors, we established that these cells became senescent a few days after radiation exposure. Senescence and SASP gene upregulation were also seen in primary PDX glioblastoma cultures differing in p53 status. Most cell lines and PDX cultures, regardless of p53 status, exhibited sustained induction of p21. p21 is documented to be upregulated by mechanisms that do not involve p53 such as activation of NF-κB (Basile et al, 2003; Nicolae et al, 2018) which might explain why p21 upregulation was seen even in cells with mutant p53. These results indicated to us that most GBM cells have the potential to undergo senescence after radiation exposure which necessitated an exploration of the underlying mechanisms and the deleterious consequences of the SASP in the context of GBM.

SASP gene expression is largely dependent upon the NF-κB pathway (Chien et al, 2011; Ohanna et al, 2011) which is activated in senescent cells by persistent DNA damage or cytosolic dsDNA in a PARP1/ATM- or cGAS/STING-dependent manner (Dou et al, 2017; Dunphy et al, 2018; Kolesnichenko et al, 2021; Ohanna et al, 2011; Yang et al, 2017). A key target of the NF-κB pathway is the pro-inflammatory cytokine IL-6 which activates the JAK-STAT3 pathway which has clear tumor promoting roles (Johnson O'Keefe and Grandis, 2018), and that also maintains constitutive NF-κB activity in tumors in a feed forward loop (Lee et al, 2009). Analysis of CM from senescent GBM cells revealed significantly high levels of a common sub-set of 24 SASP factors which included IL-6. TRRUST analyses of the secreted factors revealed activation of the STAT3 and NF-κB transcription factors as two of the most probable outcomes. Accordingly, we found that CM-SEN could activate the JAK-STAT3 and NF-κB pathways in naive GBM cells and stimulate proliferation of these cells. Activation of the NF-κB pathway in naive cells was presumably responsible for the observed upregulation of SASP genes in these cells. Such spreading of the SASP from senescent to naive cells is very similar to "senescence spreading" from damaged to undamaged cells that has been previously reported (Acosta et al, 2013; Nelson et al, 2012; Vicente et al, 2016; Xu et al, 2018; Yousefzadeh et al, 2021). It is therefore not difficult to envisage how senescent GBM cells arising after radiotherapy would not only secrete tumor promoting factors but also amplify the SASP response by inducing SASP genes in non-senescent cells, with the amplified SASP *milieu* then spurring tumor recurrence. Our findings of radiation-induced senescence in GBM and their importance are validated by a recent paper which reported that a small fraction of tumor cells in untreated primary human GBM display markers of senescence (Salam et al, 2023). Though small in number, these cells are presumably tumor promoting, as genetic ablation of such senescent cells marginally improved survival in mouse GBM models. Therefore, it is quite plausible that the much greater degree of senescence in irradiated GBM could profoundly affect recurrent tumor growth and morbidity in patients.

Senescent cells rely upon pro-survival signaling cascades and senescence-associated anti-apoptotic proteins (SCAPs) for their long-term survival, which could also be the proverbial "Achilles' heel" of these cells (Gorgoulis et al, 2019; Zhu et al, 2015). Hence, there is burgeoning interest in the aging and cancer fields in the development of "senolytic" drugs to specifically eliminate senescent cells (Chaib Tchkonia and Kirkland, 2022). The first senolytic drugs were identified based upon the hypothesis that senescent cells would be more sensitive than non-senescent cells to inhibition of these pro-survival networks or SCAPs, as exemplified by the dasatinib (D) and quercetin (Q) combination or by BH3 mimetics targeting the BCL-2 family of anti-apoptotic proteins, respectively (Birch and Gil, 2020; Zhu et al, 2015). Thus far, D + Q, fisetin (Zhu

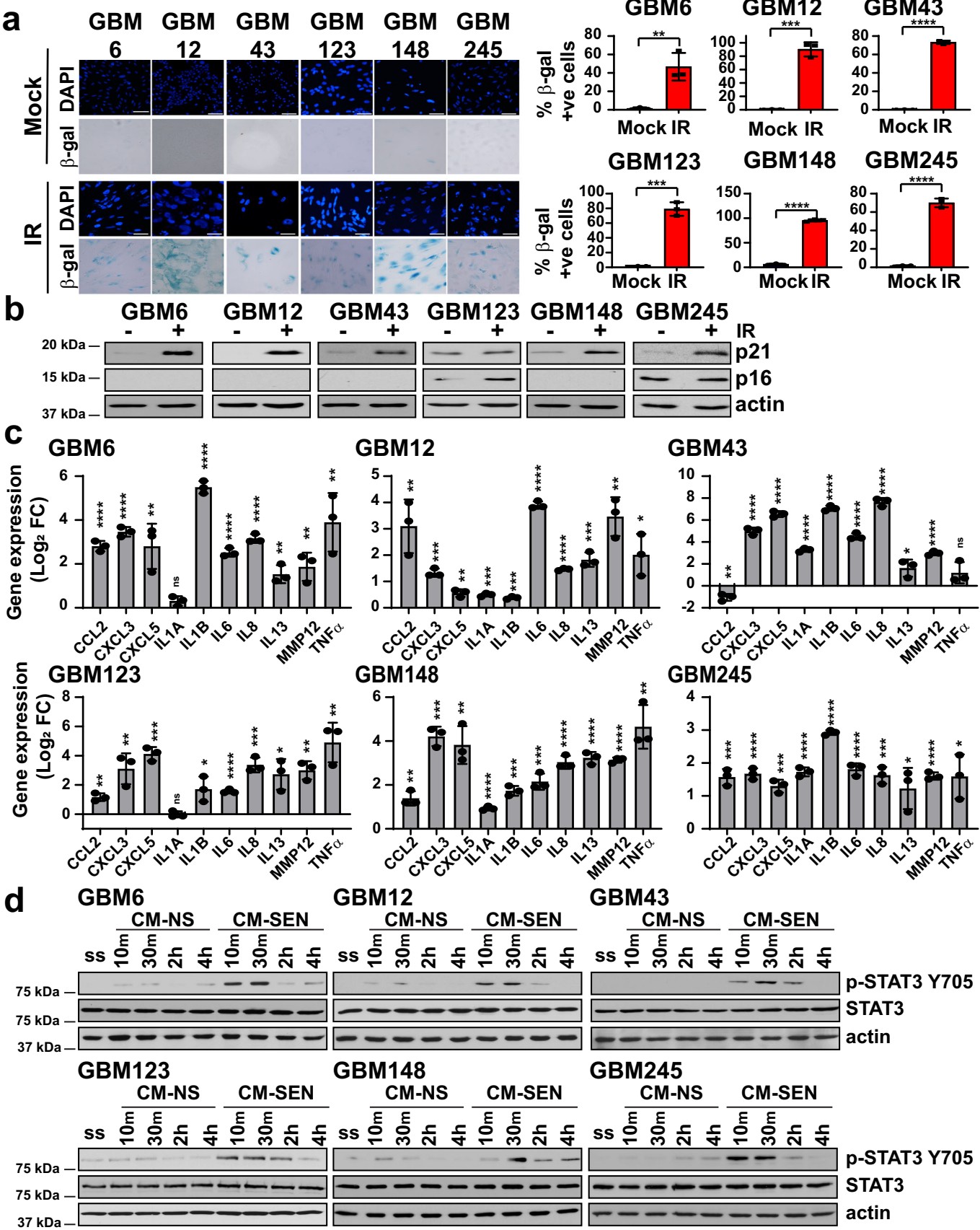

◀ **Figure 5. Senescence, upregulation of SASP genes, and paracrine effects in GBM PDX cultures.**

(A) Representative images of SA-β-gal staining of GBM PDX cultures mock-irradiated (Mock) or irradiated (IR) with 10 Gy of X-rays and then allowed to recover for 10 days ($n = 3$ with at least 100 nuclei scored for each replicate). Nuclei are stained with DAPI (blue). Plots show mean percentages of SA-β-gal-positive cells $+/-$ SD. A two-tailed Student's $t$ test was performed; GBM6 $P = 0.0064832$, GBM12 $P = 0.0001251$, GBM43 $P = 0.0000002$, GBM123 $P = 0.0001289$, GBM148 $P = 0.0000007$, GBM245 $P = 0.0000155$. Scale bar, 50 μm. (B) Whole cell extracts from mock-irradiated or irradiated PDX cultures were western blotted with anti-p21 and anti-p16 antibodies, as indicated. Actin serves as loading control. (C) Total RNA was isolated from PDX cultures 10 days after irradiation with 10 Gy of X-rays, and expression of SASP genes assessed by qRT-PCR ($n = 3$ biological replicates comprising three technical replicates each). Plots show mean fold change (Log$_2$ FC) in gene expression $+/-$ SD of SASP-related genes in irradiated GBM cells relative to mock-irradiated cells. A two-tailed Student's $t$ test was performed; ns, not significant; *$P < 0.05$; **$P < 0.01$; ***$P < 0.001$; ****$P < 0.0001$ (Please refer to Appendix Table S4 for the exact $P$ values). (D) Naive PDX cultures were exposed to conditioned media from senescent (CM-SEN) or non-senescent (CM-NS) PDX cultures for the indicated times, and activation of the JAK-STAT3 pathway assessed by western blotting with anti-phopho-STAT3 (Y705) antibody. Actin serves as loading control. Recipient cells were serum starved (ss) before addition of CM. Source data are available online for this figure.

et al, 2017), and BCL-2 family inhibitors such as navitoclax are the most frequently used senolytics in vivo. Apart from the anti-apoptotic members of the BCL-2 family, another group of proteins belonging to the inhibitor of apoptosis (IAP) family also prevent cell death by directly blocking caspase activation or by promoting pro-survival signaling (Carneiro and El-Deiry, 2020; Fulda and Vucic, 2012; Singh Letai and Sarosiek, 2019). This family of proteins has not been evaluated from the standpoint of senolytic drug discovery. Interestingly, we found that senescent GBM cells significantly upregulate the IAP family member, cIAP2, though not the related cIAP1. cIAP1 and cIAP2 both promote cancer cell survival by constitutively ubiquitinating the RIP1 adaptor protein (Bertrand et al, 2008; Beug et al, 2012). Ubiquitinated RIP1 promotes cell survival by binding to the pro-survival kinase TAK1. Inhibition of cIAP1 and cIAP2 results in a switch from cell survival to cell death as RIP1, which is no longer ubiquitinated, binds instead to caspase-8 to induce apoptosis. The upregulation of cIAP2 in senescent GBM cells could be blocked by an inhibitor of the NF-κB pathway, in keeping with the well-established role of NF-κB in inducing *BIRC3* gene expression (Karin and Lin, 2002). Concerningly, CM-SEN could induce *BIRC3* in naive GBM cells, implying that senescent cells arising after GBM radiotherapy can promote therapy resistance in their non-senescent counterparts. Indeed, we found that CM-SEN augmented the radioresistance of naive GBM cells, presumably via cIAP2 induction.

The relevance of cIAP2 to human gliomas was borne out by our perusal of the TCGA and CGGA databases wherein we observed an inverse correlation between cIAP2 expression and patient survival. Our findings of increased cIAP2 expression in senescent GBM cells indicated that SMCs that target IAPs could be effective as senolytics when given after radiation. A number of SMCs are being developed for cancer therapy (Bai Smith and Wang, 2014; Carneiro and El-Deiry, 2020; Fulda and Vucic, 2012), and the SMCs used in this study—LCL161 and birinapant—are being tested in multiple clinical trials (Morrish Brumatti and Silke, 2020). We found that both drugs could ablate cIAP2 levels in senescent GBM cells. Accordingly, we found that senescent GBM cells were strikingly more sensitive to SMCs compared to their proliferating counterparts in vitro. Intermittent dosing with the drug was necessary for complete elimination of senescent cells probably because of the transient nature of cIAP2 depletion. Importantly, we found that normal human astrocytes do not exhibit detectable levels of cIAP2 and are therefore not sensitive to SMC treatment. This observation implies that adjuvant senolytic treatment after radiotherapy could afford a therapeutic window wherein senescent GBM cells are eliminated but normal non-neoplastic brain cells are spared.

Interestingly, cIAP2 was recently identified as a potential senolytic target in a CRISPR screen (Colville et al, 2023), which validated our hypothesis-based identification of cIAP2 as a vulnerability in GBM, particularly after radiation therapy.

Informed and encouraged by the in vitro results with GBM cell lines, we re-confirmed our major findings in short-term cultures from GBM PDX lines (Vaubel et al, 2020) as a prelude to pre-clinical studies in mouse models. We found that all six PDX lines tested underwent senescence after radiation and exhibited upregulation of SASP genes, p21 or p16 expression, and, most importantly, an increase in *BIRC3* transcript and cIAP2 protein levels. CM from senescent PDX cultures stimulated proliferation of naive GBM cultures via the JAK-STAT3 pathway and also induced cIAP2 in these cells, illustrating the potential paracrine effects of senescent GBM cells on their non-senescent counterparts. Birinapant could ablate cIAP2 in senescent PDX cultures, and accordingly, we found that irradiated PDX cultures were all sensitive to birinapant treatment while naive cultures were not. As these PDX cultures have different genetic backgrounds, these results hinted at the possibility that senescence and cIAP2 induction may be seen in molecularly diverse human GBMs. Therefore, we embarked on pre-clinical studies to see if SMCs could thwart GBM recurrence when administered as an adjuvant after radiotherapy. We chose birinapant for these mouse studies as it has a good safety profile in clinical trials (Amaravadi et al, 2015; Morrish Brumatti and Silke, 2020), can cross the blood-brain-barrier, and was shown to sensitize tumor cells to immune checkpoint inhibitors in mouse models of GBM (Beug et al, 2017). We found that though orthotopic GBM12 tumors in nude mice responded to radiation treatment, the tumors inevitably recurred very quickly. On the other hand, birinapant, administered every other day, starting at 5 days post-IR, significantly blunted tumor growth in at least 50% of mice, and durable tumor control was observed in more than 30% of the treated mice. There was little or no improvement in overall survival when the mice were treated with either modality alone, while combinatorial treatment significantly prolonged survival of tumor bearing mice. It is important to point out that birinapant did not show any efficacy as a single agent and was effective only when administered as an adjuvant given after irradiation. As senescence and SASP have been reported to be both immunosuppressive and immunostimulatory in different studies and contexts (Rao and Jackson, 2016), it was critical for us to confirm that the elimination of senescent tumor cells would also be beneficial in immunocompetent mice. We found that the survival of tumor-bearing immunocompetent C57BL/6J mice could be significantly prolonged by adjuvant birinapant therapy, validating findings from the

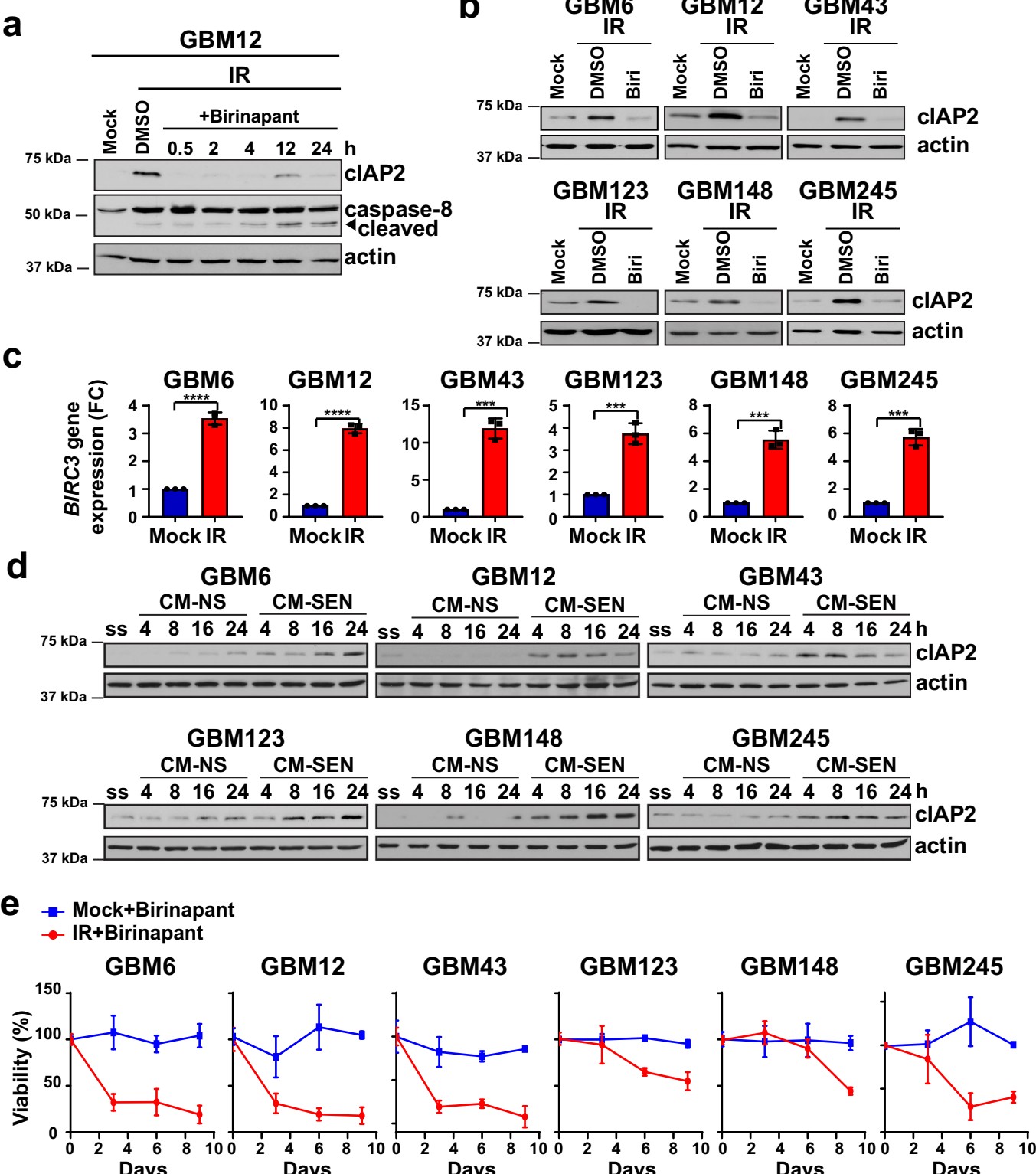

◄ **Figure 6. cIAP2 induction and birinapant sensitivity in GBM PDX cultures.**

(A) GBM12 cells were irradiated with 10 Gy of X-rays and treated with the cIAP2 inhibitor birinapant (or DMSO as control) after 10 days for the indicated times, and cIAP2 levels and cleavage of caspase-8 were assessed by western blotting. Actin serves as loading control. (B) Whole cell extracts from mock-irradiated or irradiated GBM PDX cultures treated with DMSO or birinapant for 2 h were western blotted with anti-CIAP2 antibody. (C) Plots show mean relative expression of $BIRC3 +/-$ SD in mock-irradiated (Mock) vs. irradiated (IR) PDX cultures 10 days after exposure to 10 Gy of X-rays, as assessed by qRT-PCR ($n = 3$ biological replicates comprising three technical replicates). A two-tailed Student's $t$ test was performed; GBM6 $P = 0.00004$, GBM12 $P = 0.00001$, GBM43 $P = 0.00015$, GBM123 $P = 0.00053$, GBM148 $P = 0.00026$, GBM245 $P = 0.00016$. (D) Naive PDX cultures were exposed to conditioned media from senescent (CM-SEN) or non-senescent (CM-NS) cultures for the indicated times, and expression of cIAP2 assessed by western blotting. (E) Mock-irradiated or irradiated PDX cultures were treated with birinapant or DMSO as control ($n = 3$–5 replicates per PDX) and viability was quantified by the MTT assay (normalized to that of DMSO-treated cells). The drug was replaced every 72 h. Plots show mean viability $+/-$ SD. Source data are available online for this figure.

immunodeficient mouse model. The significantly delayed recurrence of irradiated tumors treated with birinapant offers hope that should these tumors be treated with repeat irradiation or chemotherapy after relapse followed by additional rounds of birinapant, overall survival may be further extended, a premise that will be tested in future studies.

Results from our exploratory pre-clinical study are encouraging and warrant further exploration of senolytic strategies for GBM. Noted below are reasons why a well thought out clinical strategy, based upon pre-clinical studies in mouse models, may help improve therapeutic control of these highly recalcitrant tumors. A potent but specific senolytic that targets a protein uniformly expressed in GBMs will need to be administered only intermittently and will likely have fewer side effects, as seen in our mouse studies with birinapant. The administration of such a senolytic could perhaps be further guided by confirmation of induction of the target protein, such as cIAP2, after irradiation of organoids derived from primary patient samples (Sundar et al, 2022). The timing of administration could, in principle, be guided by an increase in circulating senescence markers in the patient after irradiation (Schafer et al, 2020), thereby reducing unnecessary dosing and increasing efficacy. In addition, CSF-based biomarkers may be specifically valuable for disease and SASP monitoring during senolytic therapy for gliomas (Riviere-Cazaux et al, 2023a; Riviere-Cazaux et al, 2023b): the relatively short timeframe required for senolytic ablation may also lend itself well to early phase neurosurgical studies leveraging real-time pharmacodynamic feedback from the live human tumor in situ (Riviere-Cazaux et al, 2023c). Additional anti-apoptotic proteins could be targeted in a synthetic lethal approach allowing for administration of lower doses of senolytics, thereby reducing side effects. It is important to note in this context that the results of a recently published CRISPR screen indicated that BH3 and SMAC mimetics could synergize in killing cancer cells (Colville et al, 2023). GBM cells have been shown in a previous study to be vulnerable to BH3 mimetics after genotoxic treatment (Rahman et al, 2022). Thus, studies exploring combinations of BH3 mimetics to target BCL-2 family proteins and SMAC mimetics to target cIAP2 are warranted. An added advantage of such an approach would be the clearance of senescent cells in the tumor microenvironment, such as astrocytes, which we have previously shown to be tumor promoting but vulnerable to BH3 mimetics (Fletcher-Sananikone et al, 2021). Typically, GBM patients are treated with temozolomide concomitantly with radiation and then as an adjuvant (Stupp et al, 2009). Temozolomide is a genotoxic agent that has been shown to induce senescence in GBM cells in culture (Aasland et al, 2019). It is possible that a combination of radiation

and temozolomide would further increase the extent of senescence and dependence on anti-apoptotic proteins, thereby rendering the tumors even more vulnerable to senolytic therapy. It would be important for us to test in future studies whether cells and tumors treated with a combination of radiation and temozolomide are more sensitive to birinapant or other senolytics. Such combinations might be particularly useful in the context of brainstem gliomas which by virtue of unresectable residual disease may represent a key population for translation of senolytic therapies (Kerezoudis et al, 2020). Finally, adjuvant senolytic therapy could also reduce the side effects of radiation therapy which include neurodegeneration and cognitive deficits (Burns et al, 2016) by clearing senescent cells and reducing neuroinflammation caused by SASP factors (Wissler Gerdes et al, 2020), thereby potentially improving quality of life in survivors. Of note, short senolytic interventions have been reported to reduce radiation-induced frailty, improve memory, and rejuvenate the hematopoietic system in whole-body irradiated mice (Chang et al, 2016; Fielder et al, 2022; Zhu et al, 2015).

In sum, our results underscore the usefulness of deploying SMAC mimetics to improve GBM therapy via direct elimination of senescent cells arising after radiotherapy along with a dampening of the tumor-promoting effects of SASP, while also perhaps alleviating some of the detrimental effects of chemotherapy and radiotherapy, a premise that we are starting to explore in clinical trials (Riviere-Cazaux et al, 2023a).

# Methods

## Reagents and tools table

| Reagent/resource | Reference or source | Identifier or catalog number |
|---|---|---|
| **Experimental models** | | |
| A172 | ATCC | CRL-1620 |
| LN229 | ATCC | CRL-2611 |
| U87 | ATCC | HTB-14 |
| U118 | ATCC | HTB-15 |
| Glioma 261 (GL261) | Division of Cancer Treatment and Diagnosis (DCTD) National Cancer Institute (NCI) National Institutes of Health (NIH) | https://dtp.cancer.gov/repositories/DCTDTumorRepository |
| GBM6 | Mayo Clinic Shared Services | PMID: 31852831 |
| GBM12 | Mayo Clinic Shared Services | PMID: 31852831 |
| GBM43 | Mayo Clinic Shared Services | PMID: 31852831 |
| GBM123 | Mayo Clinic Shared Services | PMID: 31852831 |
| GBM148 | Mayo Clinic Shared Services | PMID: 31852831 |

| Reagent/resource | Reference or source | Identifier or catalog number |
|---|---|---|
| GBM245 | Mayo Clinic Shared Services | PMID: 31852831 |
| Primary normal human astrocytes | iXCells Biotechnologies | 10HU-035 |
| Mouse: Athymic nude female mice homozygous for Foxn1$^{nu}$ | The Jackson Laboratory | 007850 |
| Mouse: female C57BL/6J mice | The Jackson Laboratory | 000664 |
| **Antibodies** | | |
| Anti-Lamin B1 Rabbit pAb | Abcam | ab16048 |
| Anti-Ki67 Rabbit pAb | Abcam | ab15580 |
| Anti-BrdU Purified Mouse mAb | Becton Dickinson | 347580 |
| Anti-p21 Waf1/Cip1 (12D1) Rabbit mAb | Cell Signaling Technology | 2947 |
| Anti-p16 INK4A (D7C1M) Rabbit mAb | Cell Signaling Technology | 80772 |
| HRP-conjugated Anti-Beta Actin Mouse mAb | Proteintech | HRP-66009 |
| Anti-Phospho-NF-κB p65 (Ser536) (93H1) Rabbit mAb | Cell Signaling Technology | 3033 |
| Anti-NF-κB p65 (D14E12) XP® Rabbit mAb | Cell Signaling Technology | 8242 |
| Anti-Phospho-Stat3 (Tyr705) (D3A7) XP® Rabbit mAb | Cell Signaling Technology | 9145 |
| Anti-Stat3 (D3Z2G) Rabbit mAb | Cell Signaling Technology | 12640 |
| Anti-c-IAP2 (58C7) Rabbit mAb | Cell Signaling Technology | 3130 |
| Anti-cIAP2 Rabbit pAb | Abcam | ab23423 |
| Anti-c-IAP1 (D5G9) Rabbit mAb | Cell Signaling Technology | 7065 |
| Anti-Caspase-8 (1C12) Mouse mAb | Cell Signaling Technology | 9746 |
| Anti-mouse IgG, HRP-linked Horse pAb | Cell Signaling Technology | 7076 |
| Anti-rabbit IgG, HRP-linked Goat pAb | Cell Signaling Technology | 7074 |
| Anti-Rabbit IgG (H + L) Highly Cross-Adsorbed Goat Secondary pAb, Alexa Fluor™ 488 | Invitrogen | A11034 |
| Anti-Mouse IgG (H + L) Cross-Adsorbed Goat Secondary pAb, Alexa Fluor™ 568 | Invitrogen | A11004 |
| **Oligonucleotides and other sequence-based reagents** | | |
| hGAPDH-F | GACTCATGACCACAGTCCATGC | |
| hGAPDH-R | AGAGGCAGGGATGATGTTCTG | |
| CCL2-1-F | CAGCCAGATGCAATCAATGCC | |
| CCL2-1-R | TGGAATCCTGAACCCACTTCT | |
| CXCL3-1-F | CGCCCAAACCGAAGTCATAG | |
| CXCL3-1-R | GCTCCCCTTGTTCAGTATCTTTT | |
| CXCL5-1-F | AGCTGCGTTGCGTTTGTTTAC | |
| CXCL5-1-R | TGGCGAACACTTGCAGATTAC | |
| IL1A-F | AGATGCCTGAGATACCCAAAACC | |
| IL1A-R | CCAAGCACACCCAGTAGTCT | |
| IL1B-F | ATGATGGCTTATTACAGTGGCAA | |
| IL1B-R | GTCGGAGATTCGTAGCTGGA | |
| IL6-1-F | ACTCACCTCTTCAGAACGAATTG | |

| Reagent/resource | Reference or source | Identifier or catalog number |
|---|---|---|
| IL6-1-R | CCATCTTTGGAAGGTTCAGGTTG | |
| IL8-1-F | ACTGAGAGTGATTGAGAGTGGAC | |
| IL8-1-R | AACCCTCTGCACCCAGTTTTC | |
| IL13-F | CCTCATGGCGCTTTTGTTGAC | |
| IL13-R | TCTGGTTCTGGGTGATGTTGA | |
| MMP12-1-F | GATCCAAAGGCCGTAATGTTCC | |
| MMP12-1-R | TGAATGCCACGTATGTCATCAG | |
| TNFa-2-F | AGGACCAGCTAAGAGGGAGA | |
| TNFa-2-R | TTCAGTGCTCATGGTGTCCT | |
| cIAP2-F | TTTCCGTGGCTCTTATTCAAACT | |
| cIAP2-R | GCACAGTGGTAGGAACTTCTCAT | |
| cIAP1 - F | AGCTAGTCTGGGATCCACCTC | |
| cIAP1 - R | GGGGTTAGTCCTCGATGAAG | |
| hGAPDH-F | GACTCATGACCACAGTCCATGC | |
| hGAPDH-R | AGAGGCAGGGATGATGTTCTG | |
| CCL2-1-F | CAGCCAGATGCAATCAATGCC | |
| CCL2-1-R | TGGAATCCTGAACCCACTTCT | |
| CXCL3-1-F | CGCCCAAACCGAAGTCATAG | |
| CXCL3-1-R | GCTCCCCTTGTTCAGTATCTTTT | |
| CXCL5-1-F | AGCTGCGTTGCGTTTGTTTAC | |
| CXCL5-1-R | TGGCGAACACTTGCAGATTAC | |
| IL1A-F | AGATGCCTGAGATACCCAAAACC | |
| IL1A-R | CCAAGCACACCCAGTAGTCT | |
| IL1B-F | ATGATGGCTTATTACAGTGGCAA | |
| IL1B-R | GTCGGAGATTCGTAGCTGGA | |
| IL6-1-F | ACTCACCTCTTCAGAACGAATTG | |
| IL6-1-R | CCATCTTTGGAAGGTTCAGGTTG | |
| IL8-1-F | ACTGAGAGTGATTGAGAGTGGAC | |
| IL8-1-R | AACCCTCTGCACCCAGTTTTC | |
| IL13-F | CCTCATGGCGCTTTTGTTGAC | |
| IL13-R | TCTGGTTCTGGGTGATGTTGA | |
| MMP12-1-F | GATCCAAAGGCCGTAATGTTCC | |
| MMP12-1-R | TGAATGCCACGTATGTCATCAG | |
| TNFa-2-F | AGGACCAGCTAAGAGGGAGA | |
| TNFa-2-R | TTCAGTGCTCATGGTGTCCT | |
| cIAP2-F | TTTCCGTGGCTCTTATTCAAACT | |
| cIAP2-R | GCACAGTGGTAGGAACTTCTCAT | |
| cIAP1 - F | AGCTAGTCTGGGATCCACCTC | |
| cIAP1 - R | GGGGTTAGTCCTCGATGAAG | |
| mGAPDH-F | GGCTCATGACCACAGTCCATGC | |
| mGAPDH-R | GGATGCAGGGATGATGTTCTG | |
| m-ccl2-F | ACCTGCTGCTACTCATTCACC | |
| m-ccl2-R | CATTCCTTCTTGGGGTCAGCA | |
| m-Cxcl3-F | GCACCCAGACAGAAGTCATAG | |
| m-Cxcl3-R | ACTTGCCGCTCTTCAGTATC | |
| m-Cxcl5-F | CTGGCATTTCTGTTGCTGTTC | |
| m-Cxcl5-R | TCACCTCCAAATTAGCGATCAA | |
| m-IL1A-F | CTCTGAGAACCTCTGAAACGTC | |
| m-IL1A-R | GAAACTCAGCCGTCTCTTCTT | |
| m-IL1B-F | TGCCACCTTTTGACAGTGATGA | |
| m-IL1B-R | TGTGCTGCTGCGAGATTTGA | |
| m-IL-6-F | CCACTTCACAAGTCGGAGGC | |

| Reagent/resource | Reference or source | Identifier or catalog number |
| --- | --- | --- |
| m-IL-6-R | TCTGCAAGTGCATCATCGTTGT | |
| m-IL-8-F | GCTCCATGGGTGAAGGCTAC | |
| m-IL-8-R | ACAGAAGCTTCATTGCCGGT | |
| m-IL-13-F | GCAGCATGGTATGGGAGTGT | |
| m-IL-13-R | TATCCTCTGGGTCCTGTAGATG | |
| m-MMP12-F | GGCTGCAGCATTCCAATAATC | |
| m-MMP12-R | CCATAGAGGGACTGAATGTTACG | |
| m-TNFa-F | GGCGGTGCCTATGTCTCAG | |
| m-TNFa-R | TGGTTTGTGAGTGTGAGGGTC | |
| mcIAP2-1-F | CTGGCTATTTCAGTGGCTCTTA | |
| mcIAP2-1-R | TGCAAAGTGGTAGGGACTTG | |
| **siRNAs** | | |
| MISSION® siRNA Universal Negative Control #1 | Sigma-Aldrich | SIC001-10NMOL |
| ON-TARGETplus Human BIRC3 siRNA (SMARTPool) | Horizon Discovery | L-004099-00-0005 |
| c-IAP2 siRNA (h) | Santa Cruz Biotechnology, Inc | sc-29850 |
| **Chemicals, enzymes and other reagents** | | |
| DMEM, high glucose, pyruvate | Gibco™ | 11995065 |
| DPBS, no calcium, no magnesium | Gibco™ | 14190144 |
| Gibco StemPro NSC SFM medium | Gibco™ | A1050901 |
| L-Glutamine (200 mM) | Gibco™ | A2916801 |
| Cytiva HyClone™ Non Essential Amino Acids (NEAA) 100X Solution | Thermo Fisher Scientific | SH30238.01 |
| Gibco™ Trypsin-EDTA (0.25%), phenol red | Gibco™ | 25200056 |
| StemPro™ Accutase™ Cell Dissociation Reagent | Gibco™ | A1110501 |
| Regular Fetal Bovine Serum | Corning™ | 35010CV |
| Penicillin-Streptomycin Solution | Corning™ | 30002CI |
| Astrocyte Medium | iXCells Biotechnologies | MD-0039-500ML |
| BMS-345541 | MedChemExpress | HY-10519 |
| Ruxolitinib | Selleck Chemicals | S1378 |
| Dimethyl sulfoxide (DMSO) | Sigma-Aldrich | D8418 |
| Birinapant | Selleck Chemicals | S7015 |
| LCL161 | Selleck Chemicals | S7009 |
| X-Gal (5-bromo-4-chloro-3-indolyl-β-D-galactosidase) | Thermo Fisher Scientific | FERR0404 |
| Citric acid | Thermo Scientific Chemicals | A1039530 |
| Sodium phosphate dibasic | Sigma-Aldrich | S7907-500G |
| Paraformaldehyde (PFA) | Sigma-Aldrich | P6148 |
| Potassium ferricyanide | Thermo Scientific Chemicals | 223111000 |
| Potassium Ferrocyanide Trihydrate | MP Biomedicals, Inc | 0215256080 |
| Sodium chloride (NaCl) | Sigma-Aldrich | S3014-1KG |
| MgCl2 (1 M) | Invitrogen™ | AM9530G |
| VECTASHIELD® PLUS Antifade Mounting Medium with DAPI (H-2000) | Vector Laboratories | H-2000-10 |

| Reagent/resource | Reference or source | Identifier or catalog number |
| --- | --- | --- |
| Sucrose | Sigma-Aldrich | S0389-1KG |
| Tissue-Tek O.C.T. Compound | Sakura | 4583 |
| DAPI ready made solution | Sigma-Aldrich | MBD0015 |
| FluorSave Reagent | Sigma-Aldrich | 345789 |
| Triton™ X-100 | Sigma-Aldrich | T8787-250ML |
| TWEEN® 20 | Sigma-Aldrich | P7949-500ML |
| Bovine Serum Albumin (BSA) | Sigma-Aldrich | A3059-50G |
| Tris Base Ultra Pure [Tris (Hydroxymethyl) Aminomethane] | Research Products International Corp | T60040-1000.0 |
| UltraPure™ 0.5 M EDTA, pH 8.0 | Invitrogen | 15575020 |
| Glycerol | Sigma-Aldrich | G5516-1L |
| Sodium dodecyl sulfate (SDS) | Sigma-Aldrich | L3771-500G |
| DTT [DL-Dithiothreitol] (Cleland's Reagent) | Research Products International Corp | D110005.0 |
| Ammonium Persulfate (APS) | Bio-Rad | 1610700 |
| TEMED | Bio-Rad | 1610800 |
| Ponceau S | Sigma-Aldrich | P3504-10G |
| RNeasy Mini Kit | Qiagen | 74104 |
| RNase-free DNase set | Qiagen | 79254 |
| TRI Reagent | Invitrogen | AM9738 |
| Chloroform | Sigma-Aldrich | C2432-500ML |
| Ethanol, 200 proof (100%), USP | Decon™ Laboratories, Inc | 2701 |
| High-Capacity RNA-to-cDNA Kit | Thermo Fisher Scientific | 4387406 |
| PowerUp SYBR Green Master Mix | Thermo Fisher Scientific | A25777 |
| BrdU | Sigma-Aldrich | B5002 |
| Hydrochloric Acid (HCl) | Supelco | HX0603-4 |
| Crystal Violet | Sigma-Aldrich | C6158 |
| PROTOCOL™ 10% Buffered Formalin | Fisher HealthCare™ | 23-245684 |
| MTT (3-(4,5-Dimethylthiazol-2-yl)-2,5- Diphenyltetrazolium Bromide) | Thermo Fisher Scientific | M6494 |
| Isopropanol/2-Propanol | Sigma-Aldrich | I9516-500ML |
| RNAiMAX | Invitrogen | 13778150 |
| Opti-MEM™ | Gibco | 11058021 |
| propidium iodide | Invitrogen | P1304MP |
| LIVE/DEAD™ Viability/ Cytotoxicity Kit | Invitrogen | L3224 |
| Hoechst 33342 | Invitrogen | H3570 |
| Fluriso™, Isoflurane | VetOne® | 502017 |
| Captisol-research grade | Cydex | Lot: NC-06C-21057 |
| D-luciferin | Gold Biotechnology | LUCK-1G |
| Precision Plus Protein™ Dual Color Standards | Bio-Rad | 1610394 |
| Precision Plus Protein™ All Blue Prestained Protein Standards | Bio-Rad | 1610393 |
| 10x Tris/Glycine/SDS Buffer | Bio-Rad | 1610772 |
| Resolving Gel Buffer | Bio-Rad | 1610798 |

| Reagent/resource | Reference or source | Identifier or catalog number |
|---|---|---|
| Stacking Gel Buffer | Bio-Rad | 1610799 |
| 30% Acrylamide/Bis Solution 29:1 | Bio-Rad | 1610156 |
| Can Get Signal® Immunoreaction Enhancer Solution | Cosmo Bio USA, Inc | TYBNKB101 |
| Restore™ PLUS Western Blot Stripping Buffer | ThermoFisher Scientific | 46430 |
| Methanol | Sigma-Aldrich | 179337 |
| Glycine | Sigma-Aldrich | G8898-1KG |
| IGEPAL® CA-630 | Sigma-Aldrich | I8896-100ML |
| Sodium deoxycholate | Sigma-Aldrich | D6750-25G |
| Quick Start™ Bradford 1x Dye Reagent | Bio-Rad | 5000205 |
| Pierce™ ECL Western Blotting Substrate | ThermoFisher Scientific | 32106 |
| SuperSignal™ West Pico PLUS Chemiluminescent Substrate | ThermoFisher Scientific | 34577 |
| SuperSignal™ West Femto Maximum Sensitivity Substrate | ThermoFisher Scientific | 34095 |
| **Software** | | |
| Living Image data acquisition software | Xenogen Corp. | |
| NIS-Elements AR 5.42.06 software | Nikon Instruments, Inc | |
| ImageJ Fiji | https://fiji.sc/ PMID: 22743772 | |
| FastQC (v.0.11.5) | Andrews S (2010) | |
| Trimmomatic v0.39 | PMID: 24695404 | |
| Kallisto v0.50.0 | PMID: 27043002 | |
| EdgeR | PMID: 19910308 | |
| GraphPad Software v10.2.2 | GraphPad Software, Inc | |
| Gene Set Enrichment Analysis (GSEA) | PMID: 12808457 PMID: 16199517 | |
| SDS 2.4 Software | Thermo Fisher Scientific | |
| FlowJo™ v10.10 | BD Life Sciences | |
| Cytoscape v3.10.3 | PMID: 14597658 | |
| Gen5 Microplate Reader and Imager Software v3.11 | BioTek | |

## Cell culture

GBM cell lines (A172, LN229, U87, and U118) were obtained from the American Type Culture Collection (ATCC). GL261 cells were obtained from the Division of Cancer Treatment and Diagnosis (NIH). All cells were maintained in DMEM medium supplemented with 10% fetal bovine serum and penicillin/streptomycin in a humidified atmosphere with 5% CO2. PDX lines (GBM 6, 12, 43, 123, 148 and 245) were obtained from the Mayo Clinic Shared Services (Vaubel et al, 2020). PDX lines were maintained as neurospheres in Gibco StemPro NSC SFM medium (Gibco, Cat. No. A1050901) supplemented with L-glutamine or grown as short-term adherent cultures in DMEM medium supplemented with 10% fetal bovine serum, non-essential amino acids, and penicillin/streptomycin. Primary normal human astrocytes were obtained from iXCells Biotechnologies (Cat. No: 10HU-035) and maintained in Astrocyte Medium (iXCells Biotechnologies, Cat No: MD-0039-500ML) and penicillin/streptomycin. Cell lines and PDX cultures were not authenticated as they were directly procured from ATCC, NIH and the Mayo Clinic. All cells were mycoplasma free.

## Irradiation, drug treatments and conditioned media collection

Cells were irradiated using the X-RAD320 irradiator (Precision X-Ray: 320 kV, 12.5 mA, 1.65 mm Al filter, at 50 cm) with a dose rate of 2.4 Gy/min. Cells were treated with 2.5 µM of BMS-345541 (MedChemExpress, Cat. No: HY-10519) or 1.2 µM of ruxolitinib (Selleck Chemicals, Cat. No: S1378), for 2 h to suppress the NF-κB pathway or JAK-STAT3 pathway, respectively. Cells were treated with DMSO as control. For media transfer experiments, donor cells were shifted to serum-free media for 48-72 h before collection of conditioned media. Recipient cells were shifted to serum-free media for 24 h, following which the serum-free media was replaced with conditioned media. For collection of conditioned media after drug treatment, cells were first shifted to serum-free media and then treated with DMSO or BMS-345541 for 72 h. Cells were treated with 1 µM birinapant (Selleck Chemicals, Cat. No: S7015) or LCL161 (Selleck Chemicals, Cat. No: S7009) at the indicated times to target cIAP2.

## Senescence-associated β-galactosidase staining of cells

For SA-β-Gal assay, cells seeded in glass chamber slides (Falcon, Cat. No: 354104) were stained with 5-bromo-4-chloro-3-indolyl-β-D-galactosidase (X-Gal, Thermo Fisher Scientific, Cat. No: FERR0404), as described (Debacq-Chainiaux et al, 2009). Briefly, cells were washed with PBS and fixed in 4% paraformaldehyde (PFA) dissolved in PBS solution for 10 min at room temperature. After fixation, cells were washed with PBS and incubated at 37 °C without $CO_2$ with freshly prepared senescence-associated (SA)-β-galactosidase staining solution (40 mM citrate-phosphate buffer pH 6.0, 1 mg/ml X-Gal, 5 mM potassium ferrocyanide, 5 mM potassium ferricyanide, 150 mM NaCl, and 2 mM $MgCl_2$). After overnight incubation, cells were washed twice with PBS, counterstained with DAPI and mounted. Images were captured using a Nikon EclipseTi-2 Microscope (20X objective lens) coupled to DS-Fi3 Microscope Camera using Nikon NIS-Elements software. The percentage of β-gal-positive cells was determined after scoring at least 100 nuclei from 5 random fields.

## Senescence-associated β-galactosidase staining of tumor sections

Tumor-bearing brains were harvested after perfusion with PBS, fixed in 4% paraformaldehyde overnight at 4 °C, incubated in 30% sucrose in PBS at 4 °C until the samples sank, and then embedded in Tissue-Tek O.C.T. Compound (Sakura, Cat. No: 4583). Twenty-micron thick brain sections were generated using Cryostar NX50 Cryostat, washed twice with PBS, and incubated with freshly prepared senescence-associated (SA)-β-galactosidase staining solution as described above. After 4.5 h, the sections were washed thrice with PBS, counterstained with DAPI, and mounted using FluorSave Reagent (Sigma-Aldrich, Cat No: 345789). Images were

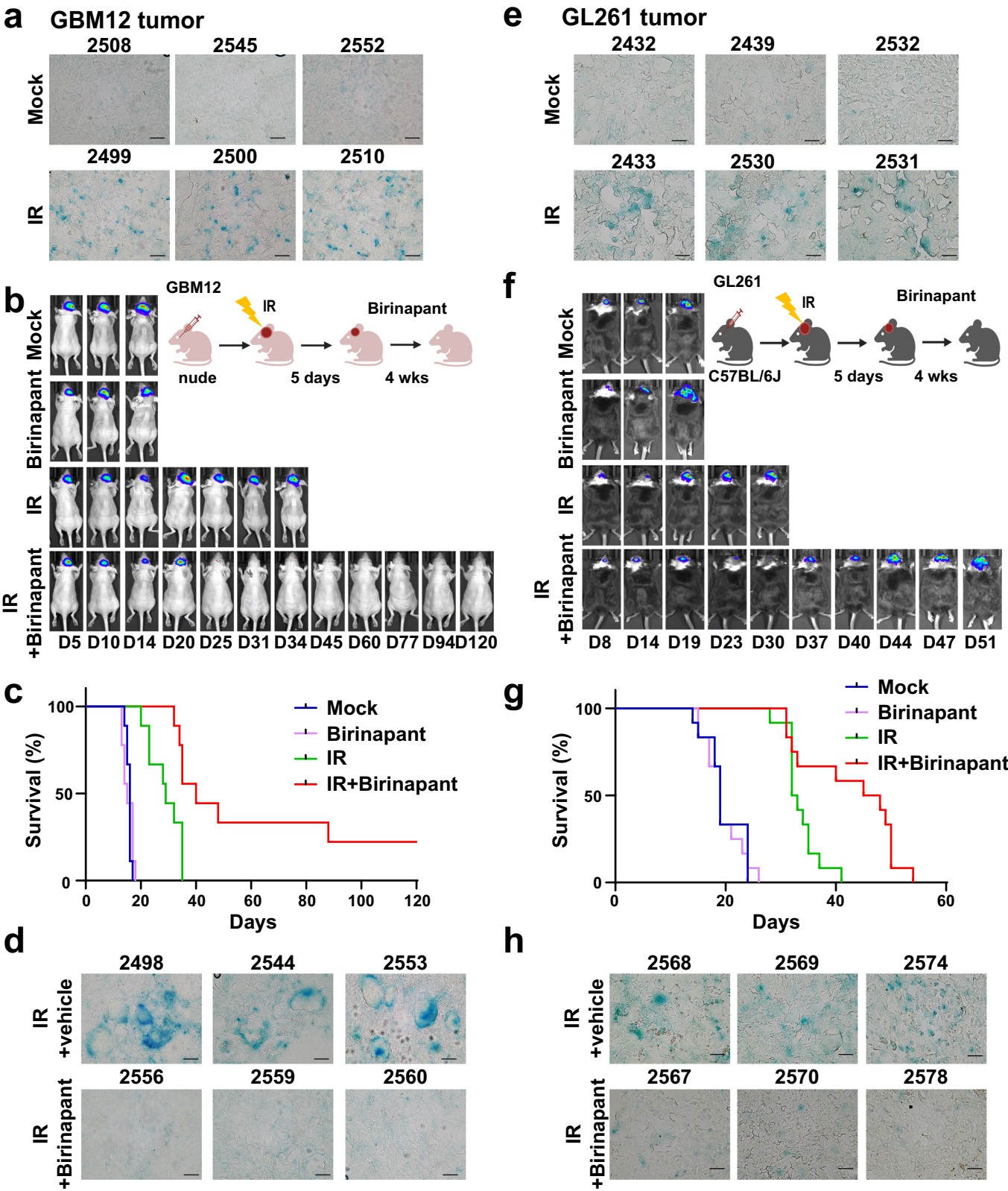

**Figure 7. Birinapant delays recurrence in pre-clinical mouse GBM models.**

(A) GBM12 cells were injected intracranially in nude mice to generate orthotopic brain tumors ($n = 3$ mice per treatment group). Mice were mock-irradiated (Mock) or cranially irradiated (IR) with X-rays (10 Gy) and sacrificed after 5 days. Representative images of SA-β-gal staining of tumor cryosections are shown (numbers denote mouse IDs). Scale bar, 20 μm. (B) GBM12 cells expressing firefly luciferase were injected intracranially in nude mice to generate orthotopic brain tumors that were monitored by BLI ($n = 9$ mice per treatment group). Mice with established tumors were randomized into the following treatment groups: (i) mock treatment (Mock), (ii) birinapant alone (Birinapant), (iii) ionizing radiation alone (IR), and (iv) IR with adjuvant birinapant (IR+Birinapat). BLI images show tumor progression in a representative mouse for each treatment arm. (C) Survival of brain tumor-bearing mice was recorded and represented in a Kaplan–Meier plot. $n = 9$ mice per group. $P = 0.004$ (IR vs IR + Birinapant). (D) Mice with GBM12 tumors ($n = 3$ mice per group) were cranially irradiated with X-rays and treated 5 days later with 10 cycles of birinapant or vehicle, as indicated, and sacrificed 24 h after the last treatment. Representative images of SA-β-gal staining of tumor cryosections are shown. Scale bar, 20 μm. (E) GL261 cells were injected intracranially in C57BL/6J mice to generate orthotopic brain tumors ($n = 3$ mice per treatment group). Mice were irradiated with X-rays (10 Gy) and sacrificed after 5 days. Representative images of SA-β-gal staining of tumor cryosections are shown. Scale bar, 20 μm. (F) GL261 cells expressing firefly luciferase were injected intracranially in C57BL/6 J mice to generate orthotopic brain tumors that were monitored by BLI ($n = 12$ mice per treatment group). Mice with established tumors were randomized into the indicated treatment groups. BLI images show tumor progression in a representative mouse for each treatment arm. (G) Survival of brain tumor-bearing mice was recorded and represented in a Kaplan–Meier plot. $n = 12$ mice per group. $P = 0.005$ (IR vs IR + Birinapant). (H) Mice with GL261 tumors ($n = 3$ mice per group) were cranially irradiated with X-rays and treated 5 days later with nine cycles of birinapant or vehicle, as indicated, and sacrificed 24 h after the last treatment. Representative images of SA-β-gal staining of tumor cryosections are shown. Scale bar, 20 μm. Source data are available online for this figure.

captured using a Nikon EclipseTi-2 Microscope (×40 objective lens) coupled to DS-Fi3 Microscope Camera using Nikon NIS-Elements software.

## Immunofluorescence staining of cells

Cells were seeded onto glass chamber slides (Falcon, Cat. No. 354104) and mock-irradiated or X-ray irradiated with a total dose of 10 Gy. Cells were fixed with 4% PFA/PBS at Day 10 post IR and permeabilized with 0.5% Triton X-100 before incubation with anti-Lamin B1 or anti-Ki67 antibodies (details in Appendix Table S3), as described before (Tomimatsu et al, 2014). Images were captured using a Zeiss LSM710 confocal microscope (×20 or ×40 objective lens). Images were generated in the Core Optical Imaging Facility, which is supported by UTHSCSA, NIH-NCI P30 CA54174 (CTRC at UTHSCSA). The percentage of Lamin B1-or Ki67-positive cells was determined after scoring at least 100 nuclei from 5 random fields.

## Western blotting and antibodies

Whole cell extracts were prepared by resuspending cell pellets in 1xSDS protein lysis buffer (67.5 mM Tris pH 6.8, 25 mM NaCl, 0.5 mM EDTA, 12.5% Glycerol, 0.25% SDS, DTT), as described before (Mukherjee et al, 2012). Excised tumors were snap frozen in liquid nitrogen and homogenized in RIPA buffer (20 mM Tris–HCl, pH 7.5, 150 mM NaCl, 1% NP-40, 1% Sodium deoxycholate, 1 mM EDTA, 1 mM EGTA, and protease and phosphatase inhibitor cocktail). Extracts were clarified by centrifugation at $16,000 \times g$ at 4 °C and the supernatant was collected and processed for western blotting, as described before (Gil del Alcazar et al, 2014). The following primary antibodies were used for western blotting: p16, p21, phosho-p65 S536, p65, phospho-STAT3 Y705, STAT3, cIAP2, cIAP1, Caspase-8, and actin. Horseradish peroxidase-conjugated secondary antibodies were from Cell Signaling Technology. Antibody dilutions and catalog numbers are provided in Appendix Table S3.

## RNA sequencing

Total RNA was extracted using RNeasy Mini Kit (Qiagen cat. No. 74104) and RNase-free DNase set (Qiagen, Cat. No: 79254), according to the manufacturer's instructions. RNA samples were run on an Agilent Tapestation 4200 to determine RNA quality, and all samples

had a RIN score 8 or higher. One microgram of total DNAse-treated RNA was then used for library preparation with the TruSeq Stranded mRNA Library Prep Kit from Illumina. Polyadenylated RNA was purified and fragmented before strand specific cDNA synthesis. cDNA was then A-tailed and indexed adapters were ligated. After adapter ligation, samples were PCR amplified and purified with AmpureXP beads, then validated on the Agilent Tapestation 4200. After normalization and pooling, sequencing was performed on a NextSeq2000 (Illumina®). Fastq files were mapped to the human reference genome hg19, and analysis was performed by quality assessment (FastQC (v.0.11.5), removal of low-quality sequences (Trimmomatic v0.39) retaining high-quality single-end reads for further alignment and reads quantification (Kallisto v0.50.0). Differential gene expression analysis used EdgeR and significantly differentially regulated genes were selected by a Benjamini–Hochberg adjusted $P$ value < 0.05 and log2-fold changes above 1.0 or below −1.0. Results were visualized with GraphPad Prism (GraphPad 10 Software) via MA and volcano plots. Gene Set Enrichment Analysis (GSEA) was performed with default settings (1000 permutations for gene sets, Signal2Noise metric for ranking genes). Senescence score was assessed using SenCan Classifier (Jochems et al, 2021) (https://rhpc.nki.nl/sites/senescence/classifier.php).

## Quantitative reverse transcription PCR

Total RNA was extracted from cells in culture and purified using TRI Reagent (Invitrogen, Cat. No: AM9738), according to the manufacturer's instructions. cDNA synthesis was performed using High-Capacity RNA-to-cDNA Kit (Thermo Fisher Scientific, Cat. No: 4387406), following the manufacturer's instructions. Quantitative PCR was performed to analyze gene expression using PowerUp SYBR Green Master Mix (Thermo Fisher Scientific, Cat. No: A25777) on an Applied Biosystems 7900HT Fast Real-Time PCR System with SDS 2.4 Software. All samples were amplified in triplicate. Measurements were standardized to *GAPDH* housekeeping gene. Expression data are presented as fold change using delta-delta Ct to measure statistical significance in the observed changes.

## Public glioma transcriptomic data analysis

Data were selected from two different databases for Lower Grade Glioma and Glioblastoma merged cohort assessment: The Cancer

Genome Atlas (TCGA) GBMLGG cohort containing 1129 cases downloaded from Firehose Broad GDAC of Broad Institute of MIT & Harvard (https://gdac.broadinstitute.org/); and the Chinese Glioma Genome Atlas (CGGA) containing 625 cases (http://www.cgga.org.cn/). Only data from patients with both transcript expression levels and clinical data available were considered for the analysis. To evaluate the correlation of cIAP2 mRNA expression levels with survival of glioma patients, a ROC curve was obtained by plotting TPM levels versus binary values of live-dead status with further calculation of the Youden Index (Fluss Faraggi and Reiser, 2005) to establish the threshold between low and high cIAP2 expression levels for survival evaluation by Kaplan–Meier assessment.

## Multiplex bead-based immunoassay

For collection of conditioned media for SASP analyses, mock-irradiated or irradiated cells were washed three times with PBS at day 7 post IR and shifted to serum-free DMEM. Conditioned media were harvested after 72 h, debris removed by centrifugation at 3000 RPM for 10 min and filtration through a 0.45 μm membrane, and flash frozen at −80 °C until analysis. Cytokines, chemokines, and growth factors in conditioned media from naive or senescent cells were quantified by Multiplex Bead-based Immunoassay, as described before (Xu et al, 2015). Human multiplex kits were purchased from R&D Systems (Bio-Techne, Minneapolis, MN) and the fluorescence readout was performed using the FlexMap 3D Instrument (Luminex) at the Facility for Geroscience Analysis (FGA), an analytical laboratory of Translational Geroscience Network (TGN). All assays were performed according to the manufacturer's protocols. More than 80% of the targets were within the detectable range (signals from all samples higher than the lowest standard), and the intraplate coefficient of variation (CV) was below 5%, and interplate CV below 15%.

## Proliferation assay

Cells were seeded onto glass chamber slides (Falcon, Cat. No: 354104) and grown overnight in serum free media, after which cells were shifted to conditioned media for 24 h. Cells were labeled with 10 μM BrdU (Sigma, Cat. No: B5002) for 4 h and fixed thereafter with 4% paraformaldehyde/PBS. Cells were incubated with 1.5 M HCl for 30 min for denaturing. Cells were immunofluorescence stained with anti-BrdU antibody (Becton Dickinson, Cat. No: 347580) after permeabilization with 0.5% Triton-X in PBS, as described (Tomimatsu et al, 2014). Images were captured using a Nikon EclipseTi-2 Microscope (20X objective lens) coupled to ORCA-Fusion Microscope Camera using Nikon NIS- Elements software. The percentage of BrdU-positive cells was determined after scoring at least 100 nuclei from 5 random fields.

## Colony formation assays

Cells were plated in triplicate onto six-well plates (LN229, 300 cells; A172, 300 cells; U118, 500 cells; U87, 1000 cells per well) and after 8 h, the medium was replaced with serum free DMEM. After overnight incubation, the medium was changed to conditioned medium (CM-NS or CM-SEN) and incubated for 8 h followed by irradiation with the indicated doses of X-rays. After 24 h, an equal

volume of DMEM with 10% FBS was added to the conditioned media. Surviving colonies were stained with crystal violet (Sigma, Cat. No: C6158) in 10% neutral buffered formalin 10 to 14 days later.

## Cell viability assessment

Cells were seeded on 96- and 24-well plates for MTT (3-(4,5-dimethylthiazol-2-yl)-2,5-diphenyltetrazolium Bromide) assay and crystal violet staining, respectively. Cells were subjected to treatment with 1 μM of Birinapant or LCL161 starting on day 10 after irradiation with 10 Gy of X-rays. Drugs were replaced every 72 h. Mock-irradiated cells were similarly drug-treated in parallel. On days 0, 3, 6 and 9 of the drug treatment regimens, cells were washed with PBS followed by staining with 0.05% crystal violet in 4% formalin/PBS for 15 min. Plates were air-dried for 24 h, and bright field images were captured on a Leica DMi1 microscope. In parallel, for the MTT assay, fresh medium containing 0.5 mg/mL MTT (Thermo Fisher Scientific, Cat. No: M6494) was added to cells and incubated for 3 h at 37 °C, followed by removal of MTT solution and solubilization of formazan crystals with acidified isopropanol solution. The absorbance of solubilized formazan was measured at 570 nm, and the absorbance values of the samples were determined by subtracting the absorbance of the blank. The percentages of viable cells were estimated by calculating the ratio between the absorbance from birinapant- or LCL161-treated groups to vehicle (DMSO)-treated cells for the respective time points.

## Propidium iodide exclusion and LIVE/DEAD™ assay after siRNA knockdown

Mock-irradiated or irradiated cells (at day 5 post-IR) were transfected with siRNAs targeting cIAP2 every 3 days, for a total of 9 days, using lipofectamine RNAiMAX (Invitrogen, Cat. No: 13778150) in Opti-MEM™ (Gibco, Cat. No: 11058021), following manufacturer's instructions. For propidium iodide exclusion (Crowley et al, 2016), cells were collected, washed in PBS, and resuspended in 50 μg/mL propidium iodide (Invitrogen, Cat. No: P1304MP) in PBS without fixation, and immediately analyzed by FACS. For each flow cytometric analysis, 10,000 events were recorded using a FACS Canto™ II flow cytometer (BD Biosciences, NJ, USA). Propidium iodide uptake was used as a readout to calculate the fraction of dead cells. Alternatively, cell death was assessed by Calcein-AM/Ethidium Homodimer-1 staining. Cells growing in 24-well plates were transfected with siRNAs as described above and incubated with 2 μM Calcein-AM and 2 μM ethidium homodimer-1 (LIVE/DEAD™ Viability/Cytotoxicity Kit, Invitrogen, Cat. No; L3224) and 5 μg/ml of Hoechst 33342 (Invitrogen, Cat. No: H3570) in serum-free DMEM for 30 min at 37 °C in 5% $CO_2$. Images were captured using a Nikon EclipseTi-2 Microscope (20X objective lens) coupled to ORCA-Fusion Microscope Camera using Nikon NIS-Elements software. The percentages of live (green staining) versus dead (red staining) cells were calculated after scoring at least 100 nuclei from five random fields for each treatment condition.

## Orthotopic tumor studies

Athymic nude female mice homozygous for Foxn1[nu] (6 weeks in age) and female C57BL/6 J mice (6 weeks in age) were purchased

from Jackson Laboratories (The Jackson Laboratory, Cat Nos: 007850; 000664). All mice were housed at the UTHSCSA-GCCRI animal facility in a standard housing condition with indoor temperature of around 22 °C and a 12 h light–dark cycle. Water bottles and cages were changed twice a week. Autoclaved food was provided ad libitum. The health of mice was regularly monitored by veterinarians as per IACUC guidelines and environmental enrichment was provided. For intra-cranial stereotactic injections, mice were anesthetized using isoflurane and mounted on an acrylic bed equipped with a nose cone in prone position. Luciferase-tagged GBM12 (Vaubel et al, 2020) or GL261 (Fletcher-Sananikone et al, 2021) cells were suspended in PBS and delivered into the right corpus striatum of the brains of athymic nude or C57BL/6 J mice, respectively (300,000 or 100,000 cells per injection for GBM12 or GL261, respectively). Tumor development was monitored by bioluminescence imaging (BLI). Five to nine days after inoculation, mice were randomized into four groups with average BLIs of approximately $6$–$9 \times 10^6$ photon/sec or $2$–$2.5 \times 10^6$ photon/sec, for athymic nude or C57BL/6J mice, respectively, for the following treatments: i) mock treatment (Mock), ii) birinapant alone (B), iii) IR alone (IR), and iv) IR with adjuvant birinapant (IR + B). Mice from the IR and IR + B groups were anesthetized and cranially irradiated with an X-ray device (X-RAD 320, Precision X-ray; 320 kV, 12.5 mA, 1.65 µm Al filter, at 43 cm) at a dose rate of 3 Gy/min with a total dose of 10 Gy, as described before (Fletcher-Sananikone et al, 2021). Mice from the Mock and B groups were anesthetized and sham irradiated. Mice from the B and IR + B groups were treated with birinapant (30 mg/kg; Selleck Chem, Cat. No: S7015) in vehicle (mixture of 15% captisol in water and 15% captisol in 0.1 M hydrochloric acid; ratio of 95.04:4.96; Captisol-research grade, Cydex, Lot: NC-06C-21057) by oral gavage every 48 h for a maximum of 15 doses, starting five days after irradiation or sham irradiation. Mice from the Mock or IR groups received only vehicle by oral gavage. Mice were sacrificed at the end of the experimental period or when they became moribund due to their tumor burden. Mice were perfused with 1× PBS followed by 4% PFA. Brains were dissected out, post-fixed by immersion in 4% PFA, and either embedded in paraffin or processed for cryosectioning. All animal studies were performed under protocols approved by the Institutional Animal Care and Use Committee of UT Health San Antonio (Animal Protocol No: 20200104AR).

## Noninvasive cranial bioluminescence imaging

Bioluminescence images of tumor-bearing mice were obtained using the IVIS Lumina System (Xenogen Corp.) coupled to Living Image data acquisition software (Xenogen Corp.), as described before (Fletcher-Sananikone et al, 2021). During imaging, mice were anaesthetized with isoflurane (Baxter International Inc.) and a solution of D-luciferin (180 mg/kg in PBS; Gold Biotechnology, Cat. No: LUCK-1G) was administered intraperitoneally. Images were acquired at 10 min post-luciferin administration and peak luminescence signals were recorded. The BLI signals emanating from the tumors were quantified by measuring total photon flux within the region of interest (ROI) using the Living Image software package.

### The paper explained

#### Problem

Glioblastomas (GBM) are lethal brain tumors which are treated with high doses of ionizing radiation (IR), yet these tumors invariably recur quickly, and the recurrent tumors are highly therapy resistant. As radiation will remain the mainstay of GBM therapy for the foreseeable future, there is an urgent unmet need to elucidate mechanisms underlying GBM recurrence after radiotherapy and devise strategies to delay or prevent recurrence.

#### Results

We report here that radiation itself counterintuitively drives recurrence by inducing senescence of tumor cells. Mechanistically, the senescent tumor cells secrete senescence-associated secretory phenotype (SASP) factors that spur recurrence and therapy resistance via activation of JAK-STAT3 and NF-κB pathways in their non-senescent counterparts. We identify a novel vulnerability in senescent GBM cells—the inhibitor of apoptosis protein 2 (cIAP2)—that can be targeted using a potent senolytic approach involving SMAC mimetics. We find that the SMAC mimetic birinapant can strikingly delay or prevent recurrence in multiple mouse GBM models by selectively eliminating senescent GBM cells arising after radiotherapy while sparing normal brain cells.

#### Impact

As senolytics are poised to become a critical therapeutic approach for a host of human diseases that are promoted by senescence, the novel senolytic described in this study could be useful not just for the development of adjuvant senotherapy to delay GBM recurrence in the clinic but also for the development of strategies to combat other SASP-driven pathologies.

## Statistical analyses

All experiments were repeated at least three times. Statistical analyses were performed by using two-tailed Student's *t* test with GraphPad Prism (GraphPad 10 Software). Error bars represent S.D. (ns, not significant; $*P < 0.05$; $**P < 0.01$; $***P < 0.001$; $****P < 0.0001$). Kaplan–Meier functions were used to illustrate survival profiles using Mantel–Cox tests. Mice were randomized into experimental groups with average BLIs of ~$6$–$9 \times 10^6$ photon/sec or $2$–$2.5 \times 10^6$ photon/sec, for athymic nude or C57BL/6J mice, respectively. All injected mice were included in the study. Except for image acquisition of tumor tissue sections, no blinding was performed.

## Data availability

The datasets produced in this study are available in the following database: RNA-Seq data: Gene Expression Omnibus GSE271928.

The source data of this paper are collected in the following database record: biostudies:S-SCDT-10_1038-S44321-025-00201-x.

## Peer review information

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

## Acknowledgements

SB is supported by grants from the NIH (RO1CA246807, RO1CA258381, and PO1CA275717-A17183)). PS is supported by an NIH grant (R35 CA241801). AH is supported by NIH grants (RO1CA244212 and R01NS119225). WZ is supported by a NIH grant (R01GM141091) and an American Cancer Society Research scholar grant (RSG-22-721675-01-DMC). JS is supported by a grant from the NIH (U01CA227954). DZ is supported by grants from the NIH (RO1CA241191 and RO1CA260239). JLK, TT, and JMEN are supported by a NIH grant (R33AG061456) and JLK and TT are supported by a NIH grant (R37AG013925) and the Connor Fund, Robert J and Theresa W Ryan, and the Noaber Foundation. TCB is supported by grants from the NIH (R37CA276851 and R61NS122096) and DOD (CA200745).

## Author contributions

**Nozomi Tomimatsu**: Data curation; Formal analysis; Investigation; Visualization. **Luis Fernando Macedo Di Cristofaro**: Conceptualization; Data curation; Formal analysis; Validation; Investigation; Writing—review and editing. **Suman Kanji**: Data curation; Formal analysis; Investigation; Methodology; Writing—review and editing. **Lorena Samentar**: Data curation; Formal analysis; Investigation; Writing—review and editing. **Benjamin Russell Jordan**: Formal analysis; Investigation. **Ralf Kittler**: Formal analysis. **Amyn A Habib**: Conceptualization; Resources; Formal analysis. **Jair Machado Espindola-Netto**: Formal analysis; Investigation; Methodology. **Tamara Tchkonia**: Formal analysis; Investigation; Writing—review and editing. **James L Kirkland**: Formal analysis; Writing—review and editing. **Terry C Burns**: Formal analysis; Writing—review and editing. **Jann N Sarkaria**: Resources; Writing—review and editing. **Andrea Gilbert**: Resources. **John R Floyd**: Resources; Writing—review and editing. **Robert Hromas**: Resources; Formal analysis; Writing—review and editing. **Weixing Zhao**: Resources. **Daohong Zhou**: Conceptualization; Formal analysis; Methodology; Writing—review and editing. **Patrick Sung**: Resources; Funding acquisition. **Bipasha Mukherjee**: Conceptualization; Formal analysis; Supervision; Investigation; Methodology; Writing—original draft; Writing—review and editing. **Sandeep Burma**: Conceptualization; Resources; Formal analysis; Supervision; Funding acquisition; Validation; Investigation; Visualization; Writing—original draft; Project administration; Writing—review and editing.

Source data underlying figure panels in this paper may have individual authorship assigned. Where available, figure panel/source data authorship is listed in the following database record: biostudies:S-SCDT-10_1038-S44321-025-00201-x.

## Disclosure and competing interests statement

The authors declare no competing interests.

# Expanded View Figures

**Figure EV1.  Senescent GBM cells secrete SASP factors that can potentially activate the JAK-STAT3 and NF-kB pathways.**

(**A**) Representative images of GBM cell lines immunofluorescence stained for Lamin B1 (green) and (**B**) Ki67 (green), 10 days after irradiation (IR) with 10 Gy of X-rays or mock-irradiation (Mock). $n = 3$ with at least 100 nuclei scored for each replicate. Nuclei are stained with DAPI (blue). Plots show mean percentages $+/-$ SD of Lamin B1- or Ki67-positive cells. A two-tailed Student's $t$ test was performed; Lamin B1 - LN229 $P = 0.00000163109$, A172 $P = 0.00000004724$, U118 $P = 0.00000000002$, U87 $P = 0.00000702223$; Ki67 - LN229 $P = 0.000030$, A172 $p = 0.000002$, U118 $p = 0.000028$, U87 $P = 0.000005$. Scale bar, 50 μm. (**C**) Heatmap of top 100 differentially expressed genes in LN229 or A172 GBM cells mock-irradiated (Mock) or irradiated (IR) with 10 Gy of X-rays and then allowed to recover for 10 days ($n = 3$), as assessed by RNA sequencing. (**D**) Senescence scores of mock-irradiated or irradiated LN229 and A172 cells ($n = 3$) generated by analysis of RNA-seq datasets using the SenCan Classifier tool. Score ranges from 0 (no senescence) to 1 (senescence). Plot shows mean senescence score $+/-$ SD for both GBM cell lines. A two-tailed Student's $t$ test was performed; LN229 $P = 0.00002368169637$, A172 $p = 0.00000000000002$. (**E**) Genes involved in SASP, Cytokine-cytokine receptor interaction, JAK-STAT signaling, and NF-kB signaling are significantly enriched in irradiated cells compared to mock-irradiated cells. Normalized Enrichment Scores (NES) are shown in the figure; $P = 0$. (**F**) SASP-transcription factor network generated by the list of 24 common cytokines secreted by all four senescent GBM cell lines cross-referenced against the TRRUST database. The visualization was generated using Cytoscape version 3.9.1. Transcription factors are shown in green. The cytokines not shown had no available information in the database. Source data are available online for this figure.

▶

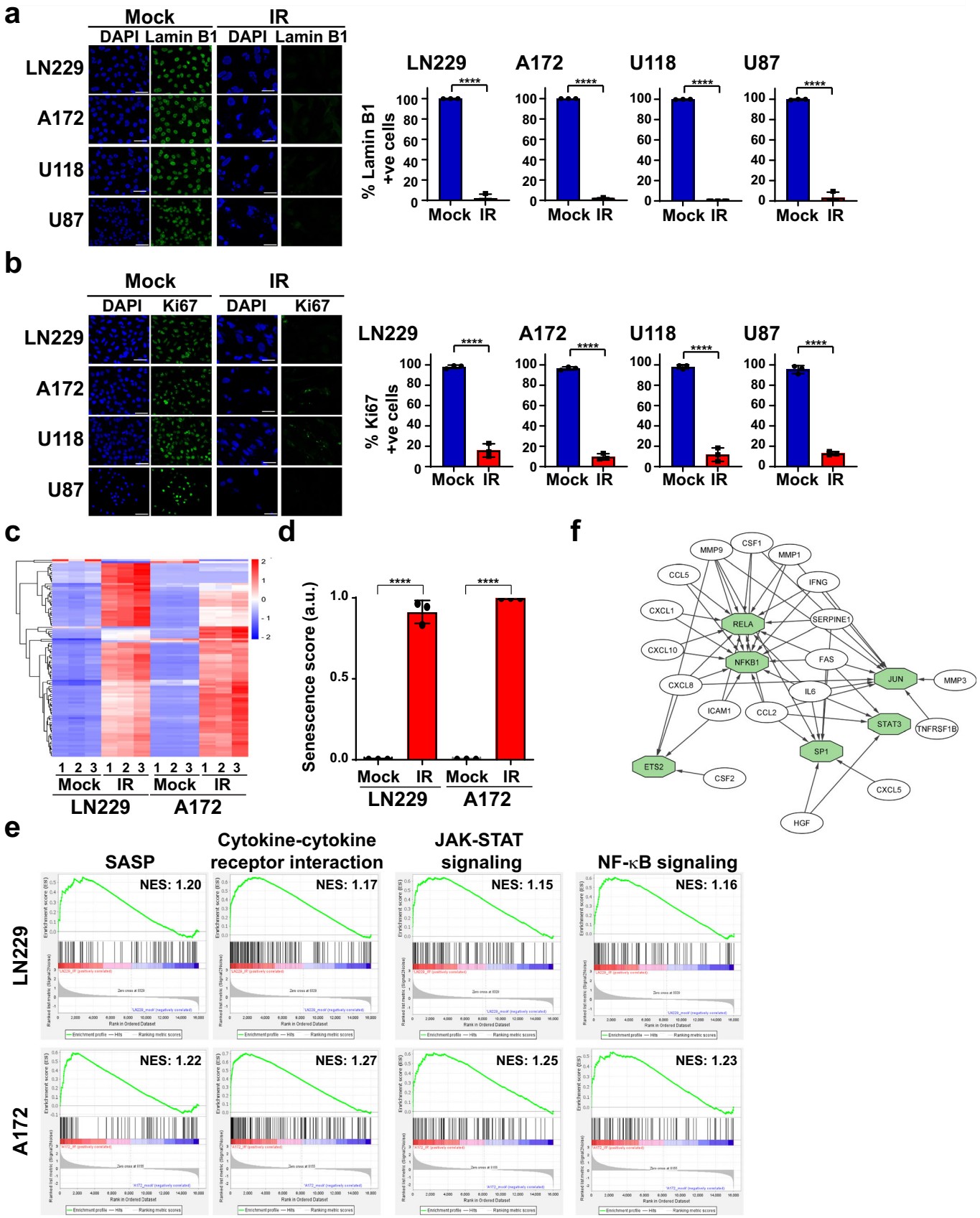

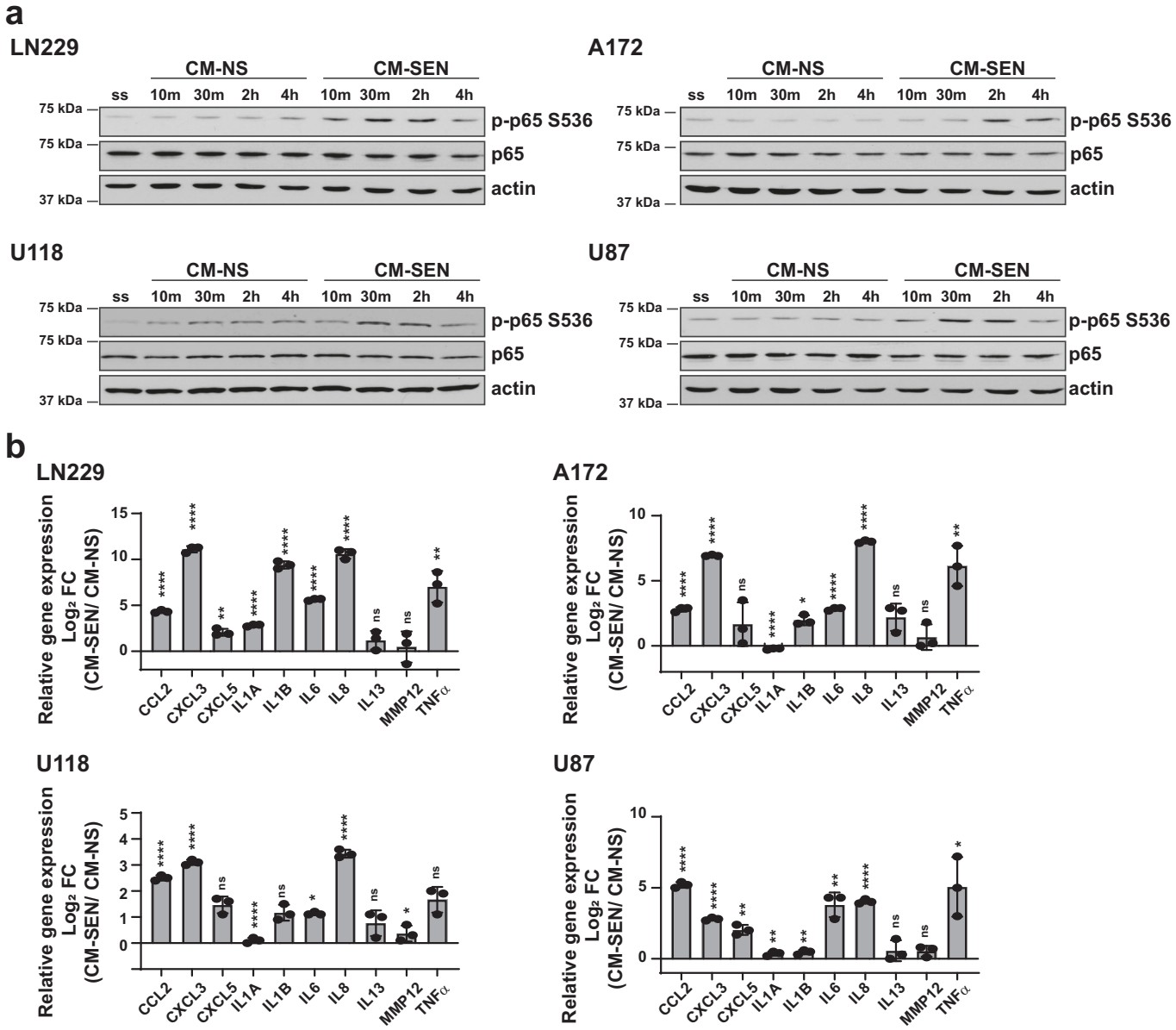

**Figure EV2. Senescent GBM cells activate the NF-kB pathway in naive GBM cells.**

(A) Naive GBM cells were exposed to conditioned media from senescent (CM-SEN) or non-senescent (CM-NS) GBM cell lines for the indicated times, and activation of the NF-kB pathway assessed by western blotting with anti-phopho-p65 (S536) antibody. Actin serves as loading control. Recipient cells were serum starved (ss) before addition of CM. (B) Serum starved GBM cells were exposed for 2 h to CM-NS or CM-SEN, and expression of SASP-related genes (relative to expression levels in serum starved cells) was assessed by qRT-PCR ($n = 3$ biological replicates comprising 3 technical replicates each). Plots show mean fold change in gene expression $+/-$ SD of SASP-related genes in CM-SEN-treated cells relative to CM-NS-treated cells. A two-tailed Student's $t$ test was performed; ns, not significant; *$P < 0.05$; **$P < 0.01$; ****$P < 0.0001$ (please refer to Appendix Table S4 for the exact $P$ values). Source data are available online for this figure.

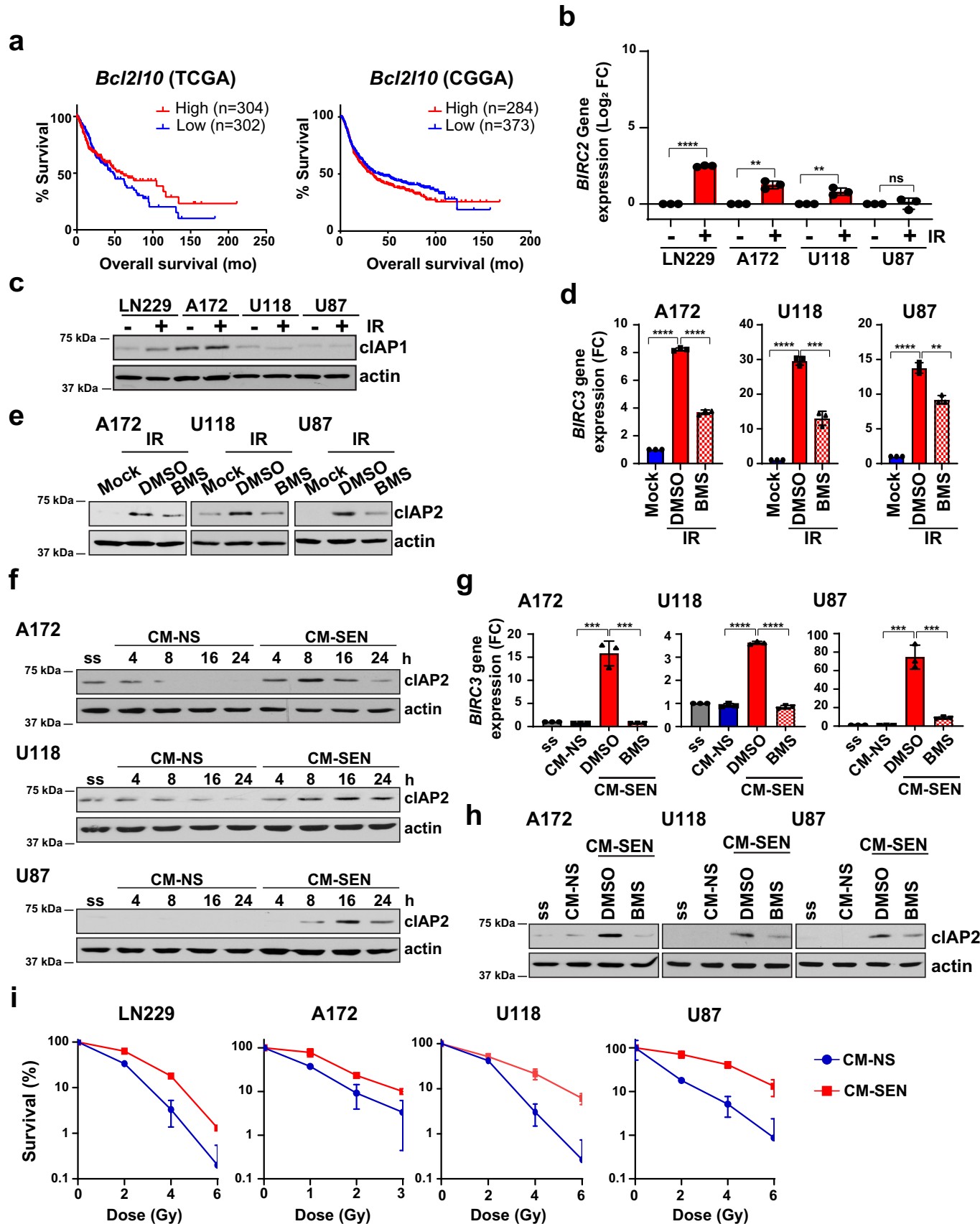

◀ **Figure EV3. Senescent GBM cells induce *BIRC3* in naive cells and promote resistance to ionizing radiation.**

(A) Kaplan–Meier curve showing lack of correlation of *Bcl2l10* expression levels with prognosis in GBMLGG patients in TCGA ($n = 606$) and CGGA ($n = 657$) cohorts as evidenced by Hazard Ratio (logrank) of 0.9692 and 0.8629 and *P* values of 0.1943 and 0.2640, respectively. (B) Plot shows mean relative expression of *BIRC2* +/− SD in mock-irradiated vs. irradiated (IR) GBM cells 10 days after exposure to 10 Gy of X-rays, as assessed by qRT-PCR ($n = 3$ biological replicates comprising 3 technical replicates each). A two-tailed Student's *t* test was performed; exact *P* values from left to right: 0.0000001, 0.0010112, 0.0044335, 0.9295618. (C) Whole cell extracts from mock-irradiated or irradiated GBM cell lines were western blotted with anti-cIAP1 antibody. Actin serves as loading control. (D) Senescent GBM cells (10 days after exposure to 10 Gy) were treated with the IKK inhibitor BMS-345541 (BMS) or DMSO as control for 72 h ($n = 3$ biological replicates comprising 3 technical replicates each), and mean relative expression of *BIRC3* +/− SD was assessed by qRT-PCR (a two-tailed Student's *t* test was performed; A172 $p = 0.00000003$, 0.00000232; U118 $p = 0.00000220$, 0.00026887; U87 $P = 0.00000818$, 0.00118914, respectively) or (E) western blotting for cIAP2. (F) Naive GBM cells were exposed to conditioned media from senescent (CM-SEN) or non-senescent (CM-NS) cells for the indicated times, and expression of cIAP2 assessed by western blotting. (G) Naive GBM cells were treated with BMS-345541 or DMSO as control for 2 h before exposure to CM-SEN ($n = 3$ biological replicates comprising 3 technical replicates each), and mean relative expression of *BIRC3* +/− SD was assessed by qRT-PCR (a two-tailed Student's *t* test was performed; A172 $P = 0.000627$, 0.000622; U118 $P = 0.000001$, 0.000001; U87 $P = 0.000593$, 0.000936, respectively) or (H) western blotting for cIAP2. (I) GBM cells were exposed to conditioned media from senescent or non-senescent cells ($n = 3$) and radiation sensitivity measured by the colony survival assay. The mean percentage of surviving colonies +/− SD (*y* axis) is plotted against the corresponding radiation dose (*x* axis). Source data are available online for this figure.

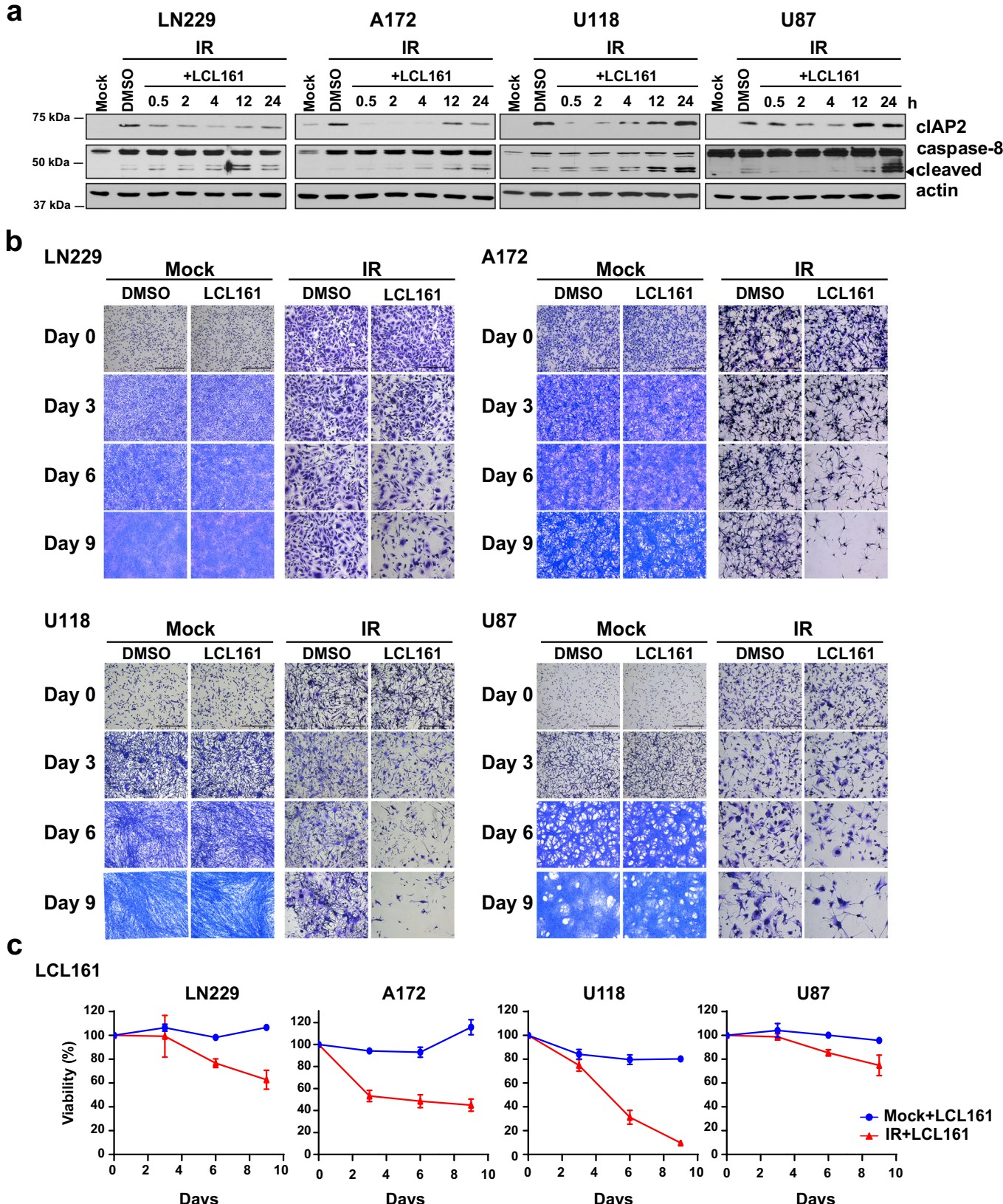

◀ **Figure EV4. The SMAC mimetic LCL161 selectively eliminates senescent GBM cells.**

(A) GBM cells were irradiated with 10 Gy of X-rays (IR) and treated with the cIAP2 inhibitor LCL161(or DMSO as control) after 10 days for the indicated times, and cIAP2 levels and cleavage of caspase-8 were assessed by western blotting. Actin serves as loading control. (B) Mock-irradiated or irradiated GBM cells were treated with LCL161or DMSO as control, and the surviving cells were visualized by staining with crystal violet at the indicated times (scale bar, 500 μm), and (C) viability (normalized to that of DMSO-treated cells) was quantified by the MTT assay ($n = 6$–9 replicates per cell line). The drug was replaced every 72 h. Plots show mean viability $+/-$ SD. Source data are available online for this figure.

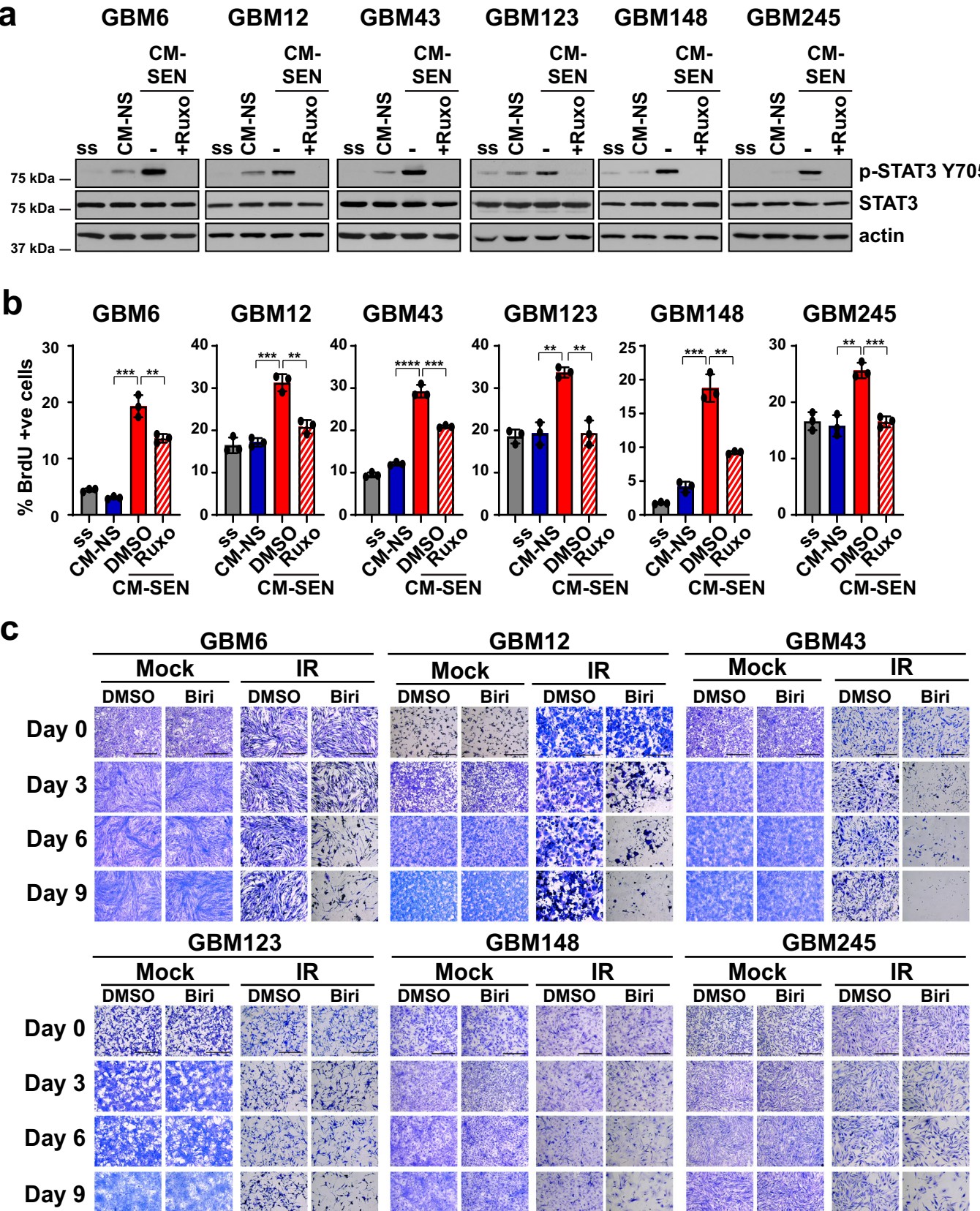

◀ **Figure EV5. Paracrine effects of senescent GBM PDX cultures and sensitivity to birinapant.**

(A) Naive GBM PDX cultures were exposed to conditioned media from senescent (CM-SEN) or non-senescent (CM-NS) PDX cultures for 30 min, and activation of the JAK-STAT3 pathway assessed by western blotting with anti-phopho-STAT3 (Y705) antibody. Actin serves as loading control. Recipient cells were serum starved (ss) before addition of CM and were untreated or treated with the JAK inhibitor ruxolitinib, as indicated. (B) Serum starved (ss) PDX cells were pulsed with BrdU after exposure to CM-NS or CM-SEN in the presence or absence of ruxolitinib ($n = 3$ with at least 100 nuclei scored for each replicate), and immunofluorescence stained with anti-BrdU antibody. Nuclei are stained with DAPI (blue). Plots show mean percentages of BrdU-positive cells $+/-$ SD. A two-tailed Student's $t$ test was performed; GBM6 $P = 0.00014$, 0.00919; GBM12 $P = 0.00040$, 0.00214; GBM43 $P = 0.00004$, 0.00069; GBM123 $P = 0.00101$, 0.00148; GBM148 $P = 0.00031$, 0.00130; GBM245 $P = 0.00191$, 0.00072, respectively. (C) Mock-irradiated or irradiated PDX cultures were treated with birinapant or DMSO as control. The drug was replaced every 72 h, and the surviving cells were visualized by staining with crystal violet at the indicated times. Scale bar, 500 μm. Source data are available online for this figure.

