## [Peer Review File · EMBO Molecular Medicine]

Targeting cIAP2 in a novel senolytic strategy prevents glioblastoma recurrence after radiotherapy

Nozomi Tomimatsu, Luis Di Cristofaro, Suman Kanji, Lorena Samentar, Benjamin Jordan, Ralf Kittler, Aryn Habib, Jair Espindola Netto, Tamara Tchkonja, Jams L. Kirkland, Terry Burns, jann sarkaria, Andrea Gilbert, John Floyd, Robert Hromas, Weixing Zhao, Daohong Zhou, Patrick Sung, Bipasha Mukherjee, and Sandeep Burma

Corresponding author(s): Sandeep Burma (burma@uthscsa.edu)

Review Timeline:

Submission Date:	9th Jul 24
Editorial Decision:	7th Aug 24
Revision Received:	18th Dec 24
Editorial Decision:	16th Jan 25
Revision Received:	28th Jan 25
Accepted:	5th Feb 25

Editor: Lise Roth

Transaction Report:

7th Aug 2024

Dear Prof. Burma,

Thank you for the submission of your manuscript to EMBO Molecular Medicine. We have now received feedback from two of the three reviewers who agreed to evaluate your manuscript. Unfortunately, referee #3 had to be withdrawn. However, given that both referees #1 and #2 provide similar recommendations, we prefer to make a decision now in order to avoid further delay in the process.

As you will see from the reports below, the reviewers both acknowledge the interest and potential clinical relevance of your study, and suggest that you address the following major points:

- Genetic inhibition of clAP2
- Use of GSC/primary GBM cells or PDX cell lines
- Additional assay(s) to measure cell viability
- Assess birinapant toxicity in vivo

If you feel you can satisfactorily address these points and others listed by the referees, you may wish to submit a revised version of your manuscript. Please attach a covering letter giving details of the way in which you have handled each of the points raised by the referees. Acceptance of the manuscript will entail a second round of review. EMBO Molecular Medicine encourages a single round of revision only and therefore, acceptance or rejection of the manuscript will depend on the completeness of your responses included in the next, final version of the manuscript. For this reason, and to save you from any frustrations in the end, I would strongly advise against returning an incomplete revision.

We are expecting your revised manuscript within three months, if you anticipate any delay, please contact us.

We require:

- 1) A .docx formatted version of the manuscript text (including legends for main figures, EV figures and tables). Please make sure that the changes are highlighted to be clearly visible.
- 2) Individual production quality figure files as .eps, .tif, .jpg (one file per figure). For guidance, download the 'Figure Guide PDF' (<https://www.embopress.org/page/journal/17574684/authorguide#figureformat>).
- 3) At EMBO Press we ask authors to provide source data for the main figures. Our source data coordinator will contact you to discuss which figure panels we would need source data for and will also provide you with helpful tips on how to upload and organize the files.
- 4) A .docx formatted letter INCLUDING the reviewers' reports and your detailed point-by-point responses to their comments. As part of the EMBO Press transparent editorial process, the point-by-point response is part of the Review Process File (RPF), which will be published alongside your paper.
- 5) A complete author checklist, which you can download from our author guidelines (<https://www.embopress.org/page/journal/17574684/authorguide#submissionofrevisions>). Please insert information in the checklist that is also reflected in the manuscript. The completed author checklist will also be part of the RPF.
- 6) All Materials and Methods need to be described in the main text using our 'Structured Methods' format, which is required for all research articles. According to this format, the Methods section includes a Reagents and Tools Table (listing key reagents, experimental models, software and relevant equipment and including their sources and relevant identifiers) followed by a Methods and Protocols section describing the methods using a step-by-step protocol format. The aim is to facilitate adoption of the methodologies across labs. More information on how to adhere to this format as well as a downloadable template (.docx) for the Reagents and Tools Table can be found in our author guidelines: <https://www.embopress.org/page/journal/17574684/authorguide#structuredmethods>
- 7) Please note that all corresponding authors are required to supply an ORCID ID for their name upon submission of a revised manuscript.

8) It is mandatory to include a 'Data Availability' section after the Materials and Methods. Before submitting your revision, primary datasets produced in this study need to be deposited in an appropriate public database, and the accession numbers and database listed under 'Data Availability'. Please remember to provide a reviewer password if the datasets are not yet public (see <https://www.embopress.org/page/journal/17574684/authorguide#dataavailability>).

9) For data quantification: please specify the name of the statistical test used to generate error bars and P values, the number (n) of independent experiments (specify technical or biological replicates) underlying each data point and the test used to calculate p-values in each figure legend. The figure legends should contain a basic description of n, P and the test applied. Graphs must include a description of the bars and the error bars (s.d., s.e.m.). Please provide exact p values.

10) Our journal encourages inclusion of *data citations in the reference list* to directly cite datasets that were re-used and obtained from public databases. Data citations in the article text are distinct from normal bibliographical citations and should directly link to the database records from which the data can be accessed. In the main text, data citations are formatted as follows: "Data ref: Smith et al, 2001" or "Data ref: NCBI Sequence Read Archive PRJNA342805, 2017". In the Reference list, data citations must be labeled with "[DATASET]". A data reference must provide the database name, accession number/identifiers and a resolvable link to the landing page from which the data can be accessed at the end of the reference. Further instructions are available at .

11) We replaced Supplementary Information with Expanded View (EV) Figures and Tables that are collapsible/expandable online. A maximum of 5 EV Figures can be typeset. EV Figures should be cited as 'Figure EV1, Figure EV2' etc... in the text and their respective legends should be included in the main text after the legends of regular figures.

12) Author contributions: CRediT has replaced the traditional author contributions section because it offers a systematic machine readable author contributions format that allows for more effective research assessment. Please remove the Authors Contributions from the manuscript and use the free text boxes beneath each contributing author's name in our system to add specific details on the author's contribution. More information is available in our guide to authors.

13) Disclosure statement and competing interests: We updated our journal's competing interests policy in January 2022 and request authors to consider both actual and perceived competing interests. Please review the policy <https://www.embopress.org/competing-interests> and update your competing interests if necessary.

14) Every published paper now includes a 'Synopsis' to further enhance discoverability. Synopses are displayed on the journal webpage and are freely accessible to all readers. They include a short stand first (maximum of 300 characters, including space) as well as 2-5 one-sentences bullet points that summarizes the paper. Please write the bullet points to summarize the key NEW findings. They should be designed to be complementary to the abstract - i.e. not repeat the same text. We encourage inclusion of key acronyms and quantitative information (maximum of 30 words / bullet point). Please use the passive voice. Please attach these in a separate file or send them by email, we will incorporate them accordingly.

15) As part of the EMBO Publications transparent editorial process initiative (see our Editorial at <http://embomolmed.embopress.org/content/2/9/329>), EMBO Molecular Medicine will publish online a Review Process File (RPF) to accompany accepted manuscripts.

In the event of acceptance, this file will be published in conjunction with your paper and will include the anonymous referee reports, your point-by-point response and all pertinent correspondence relating to the manuscript. Let us know whether you agree with the publication of the RPF and as here, if you want to remove or not any figures from it prior to publication. Please note that the Authors checklist will be published at the end of the RPF.

I look forward to receiving your revised manuscript.

Yours sincerely,

Lise Roth

***** Reviewer's comments *****

Referee #1 (Comments on Novelty/Model System for Author):

The mechanistic data are derived from established cell lines only. Validation in primary/GSC or PDX lines is recommended.

Referee #1 (Remarks for Author):

Tomitatsu et al. report that irradiation-induced senescence of tumor cells promotes glioblastoma (GBM) recurrence via a senescence-associated secretory phenotype (SASP). They show that irradiated GBM cells undergo senescence via induction of p21, upregulation of SASP genes, and secretion of SASP factors. They find that senescent GBM cells activate the JAK-STAT3 and NF- κ B pathways in naïve GBM cells to promote tumor cell proliferation and SASP spreading. Senescent GBM cells upregulate the anti-apoptotic gene BIRC3 as a survival mechanism and promote BIRC3 gene expression in naïve GBM cells. They demonstrate that SMAC mimetics can eliminate senescent GBM cells by targeting cIAP2. Targeting cIAP2 ablates radiation-induced senescence in GBM patient derived xenograft (PDX) cultures. They show that Birinapant, a new senolytic drug, can prevent recurrence in pre-clinical mouse GBM models.

This is an interesting study that shows new data with good potential translational significance. The research is mostly well performed. The manuscript is well written. However, not all findings are convincingly supported by the data.

The following issues should be addressed:

Major critique:

- The involvement of cIAP2 is demonstrated by the use of the drug Birinapant only. However, Birinapant has a stronger inhibitory effect on cIAP1 than on cIAP2 and likely also additional non-specific effects. It is therefore possible that Birinapant is acting more via cIAP1 inhibition than cIAP2. To address this issue, it is essential to use genetic manipulation of cIAP2 (si/shRNA or CRISPR) to demonstrate its involvement in the described senescence process.

- The mechanistic studies shown in Figures 1, 2, 3 and 4 were performed in established cell lines only. Key findings should also be confirmed in either GSC/primary GBM cells or PDX cell lines.

Minor critique:

- Define SMAC at first appearance.

- Showing four levels of significance with 1-4 asterisks in the figures is unnecessary, and makes the figures look crowded. Consider using just one asterisk for significance of $p < 0.05$.

Referee #2 (Comments on Novelty/Model System for Author):

The authors analyzed senescence induction using IR in GBM models and validated the potential use of SMAC mimetics (currently in clinical trials) to eliminate senescent cells.

Referee #2 (Remarks for Author):

The manuscript by Tomimatsu and colleagues analyses senescence induction using IR in GBM models and validates the potential use of SMAC mimetics (currently in clinical trials) to eliminate senescent cells. The quality of the manuscript is very high, well-written and with adequate experimental approaches, but certain aspects of the study could be improved:

- The authors use mostly MTT assays to evaluate cell viability but this assay is not optimal to analyze cytotoxicity. I would recommend to include Annexin V/PI or similar cell death analyses.
- In the in vivo experiments the potential undesired toxicities of birinapant have not been assessed. It would be informative to track the mice weight, for example, throughout the treatment.

Point by point response to referee comments

We are very grateful to the referees for their insightful comments and excellent suggestions and for their support of our glioblastoma study which has significant clinical potential. Importantly, new experiments and revisions requested by the referees have allowed us to add more mechanistic insight to the study. We have added an extra figure (and expanded view and appendix figures) to accommodate the new data generated and have made appropriate changes to the text. **New figures** generated for the revision are highlighted **in red** in our point-by-point response below. Changes to the text are highlighted using “Track Changes”. We hope that the reviewers will find the revised manuscript to be significantly improved and acceptable for publication.

Referee #1 (Remarks for Author):

This is an interesting study that shows new data with good potential translational significance. The research is mostly well performed. The manuscript is well written. However, not all findings are convincingly supported by the data.

We are grateful to Referee 1 for their support of our study which has significant translational potential for glioblastoma therapy. We have carried out the experiments suggested and made appropriate changes to improve the manuscript further.

Major critique:

- The involvement of cIAP2 is demonstrated by the use of the drug Birinapant only. However, Birinapant has a stronger inhibitory effect on cIAP1 than on cIAP2 and likely also additional non-specific effects. It is therefore possible that Birinapant is acting more via cIAP1 inhibition than cIAP2. To address this issue, it is essential to use genetic manipulation of cIAP2 (si/shRNA or CRISPR) to demonstrate its involvement in the described senescence process.

We thank the reviewer for this comment. In the original manuscript, we focused on cIAP2 because our RNA-seq data from senescent cell lines did not show significant upregulation of cIAP1 (**Fig. 3 a,b**). In the original manuscript, while we validated cIAP2 induction (**Fig. 3d,e**), we did not further evaluate cIAP1. In light of the Reviewer's concern, we have now evaluated all four GBM cells lines (**Fig. EV3b,c**) and all six GBM PDX cultures (**Appendix Fig. 3a,b**) for cIAP1 induction, and find no or low levels of cIAP1 induction after radiation, in contrast to cIAP2 (**Fig. 6b,c**).

Though birinapant has a stronger inhibitory effect on cIAP1 than on cIAP2, we did observe rapid degradation of cIAP2 in all four senescent cell lines treated with birinapant (**Fig. 4a**), and this was corroborated by the use of different SMAC mimetic, LCL161 (**Fig. EV4**). In light of the reviewer's concern, we extended these studies to all six PDX cultures (**Fig. 6b**) and to mouse tumors in vivo (**Appendix Fig. 5c**), and find that birinapant can rapidly degrade cIAP2 in all scenarios.

Based upon the reviewer's experimental suggestion, we depleted cIAP2 in senescent GBM cells using two different siRNAs and find induction of cell death upon cIAP2 depletion (**Appendix Fig. 1c-e**). Thus, we are confident that senescent GBM cells are reliant, to a large extent, on cIAP2 for survival and that birinapant exerts its effects largely via cIAP2 degradation in these cells. Moreover, in cases where there

may be redundancy between cIAP1 and cIAP2, birinapant would be very effective as a senolytic as it targets both proteins which are highly similar.

- The mechanistic studies shown in Figures 1, 2, 3 and 4 were performed in established cell lines only. Key findings should also be confirmed in either GSC/primary GBM cells or PDX cell lines.

We thank the reviewer for this suggestion. In the original manuscript, we had carried out critical studies showing induction of SASP, upregulation of cIAP2, and sensitivity to birinapant in all six PDX lines. However, we had not evaluated the paracrine effects of senescent PDX cells in conditioned media (CM) transfer experiments. Based upon the referee's suggestion, we spent considerable time and effort in generating CM from these slow growing PDX cultures. In the revised manuscript, we now show that CM from senescent PDX cultures can activate the JAK-STAT3 pathway and also induce cIAP2 in naïve PDX cells (**Fig. 5d; Fig. 6d**). This results in increased proliferation of naïve PDX cells which can be blocked by the JAK inhibitor ruxolitinib (**Fig. EV5a,b; Appendix Fig. 2**). These mechanistic studies, involving the PDX lines, re-confirm that senescent GBM cells can promote the proliferation or therapy-resistance of their non-senescent counterparts which could drive recurrence after radiotherapy.

Minor critique:

- Define SMAC at first appearance.

We thank the reviewer for pointing this out. We have expanded SMAC at first appearance.

- Showing four levels of significance with 1-4 asterisks in the figures is unnecessary, and makes the figures look crowded. Consider using just one asterisk for significance of $p < 0.05$.

We thank the reviewer for pointing this out and agree that some of the figures look busy. We respectfully ask that we be allowed to keep the 1-4 asterisk labeling, as a single star may be taken to imply a lower significance at first glance, reducing impact of the result. We have attempted, in other ways, to make the figures more streamlined, such as by removing unnecessary labels.

Referee #2 (Remarks for Author):

The manuscript by Tomimatsu and colleagues analyses senescence induction using IR in GBM models and validates the potential use of SMAC mimetics (currently in clinical trials) to eliminate senescent cells. The quality of the manuscript is very high, well-written and with adequate experimental approaches, but certain aspects of the study could be improved:

We are grateful to Referee 2 for their support of the study and are glad to note that they find our manuscript to be of high quality. We thank the reviewer for the suggestions to improve our manuscript further.

- The authors use mostly MTT assays to evaluate cell viability but this assay is not optimal to analyze cytotoxicity. I would recommend to include Annexin V/PI or similar cell death analyses.

We thank the reviewer for this suggestion. In the original manuscript, we used caspase 8 cleavage to show induction of cell death after birinapant treatment (**Fig. 4a; Fig. EV4a; Fig. 6a**), supporting the MTT

assay. We have now used two additional assays – propidium iodide exclusion and the LIVE/DEAD™ viability/cytotoxicity assay to show cell death after birinapant treatment or cIAP2 knockdown (**Appendix Fig. 1a-e**).

- In the *in vivo* experiments the potential undesired toxicities of birinapant have not been assessed. It would be informative to track the mice weight, for example, throughout the treatment.

We thank the reviewer for pointing this out. We chose birinapant for these studies as it has a good safety profile in clinical trials (Amaravadi *et al*, 2015; Morrish Brumatti & Silke, 2020) and because no overt toxicity was reported in a mouse GBM study involving combination with immune checkpoint inhibitors (Beug *et al*, 2017). We had recorded the weight of mice throughout both studies (**Fig. 7**) but had not included the data in the original manuscript. As suggested by the reviewer, we have now included the data in the revised manuscript (**Appendix Fig. 5b,k**). We find that combination treatment with radiation and birinapant is not associated with changes in body weight indicating low toxicity of the drug in animals.

16th Jan 2025

Dear Prof. Burma,

Thank you for submitting your revised study, and please accept my apologies for the delay in getting back to you during this busy time of the year. We have now received the reports from the referees who evaluated your revised manuscript. As you will see from the reports below, they are satisfied with the revisions, and I will therefore be able to accept your manuscript once the following editorial issues are addressed:

1/ Referees' comments:

Please address the remaining comments from referee #1.

2/ Manuscript text:

- Please indicate in track changes mode any new modification.
- Authors: there is a discrepancy between Jair Machado Espindola Netto in the manuscript text and Jair Machado Espindola in the submission system, please correct.
- We can accommodate a maximum of 5 keywords, please adjust accordingly.
- Methods:
 - o Thank you for providing a Reagents and tools table. Please remove it from the manuscript file and upload it as a separate file.
 - o Cell culture: please indicate whether the cells were authenticated.
 - o Mice: please provide housing and husbandry conditions
 - o Statistical analysis: please provide a statement on exclusion/inclusion criteria, blinding and randomization.
- Data Availability: Thank you for depositing your datasets. Please note that they must be made public before acceptance of the manuscript.
- Acknowledgements: Please note that the information provided in the manuscript and the submission system should match, please adjust accordingly.
- Please add a heading: "Disclosure statement and competing interests". Please review our updated policy <https://www.embopress.org/competing-interests> and update your competing interests if necessary.

3/ Figures and Appendix:

- During our standard figure check, we noticed anomalies in Figure EV5C. Please carefully check this figure panel composition, clarify and update if needed.
- Please rename the Appendix tables "Appendix Table S1" etc. and update the callouts in the manuscript text accordingly. Please upload the final version as a PDF document.
- Please address the queries from our copy editors in the figure legends:
 1. Please note that the exact p values are not provided in the legends of figures 1A, D, E; 2C; 3D, F, I; 5A, C; 6C; EV1 A, B, D; EV2 B; EV3 B, D, G; EV5 B
 2. Please indicate the statistical test used for data analysis in the legends of figures 1A, C, D, E; 2C; 3D, F, I; 5A, C; 6C; EV1 A, B, D; EV2 B; EV3 B, D, G; EV5 B.
 3. Although 'n' is provided, please describe the nature of entity for 'n' in the legends of figures 1A, C, D, E; 2C, 3B, D, F, I; 4C; 5A, C; 6C, E; EV1 A, B, D; EV2 B; EV3 B, D, G, I; EV4 C, EV5 B
 4. Please note that the measure of center for the error bars needs to be defined in the legends of figures 1A, D, E; 2C, 3B, D, F, I; 4C; 5A, C; 6C, E; EV1 A, B, D; EV2 B, EV3 B, D, G, I; EV4 C, EV5 B.

4/ Source Data: Thank you for providing detailed Source Data. We would appreciate if you could submit the numeric files as excel documents, not prism files.

5/ Checklist:

- Experimental animals, housing and husbandry conditions: please fill in.
- Experimental study design and statistics: please fill in the sections on blinding and inclusion/exclusion criteria.

6/ Synopsis:

- I introduced minor edits in your synopsis, please let me know if you agree or amend as you see fit:

"Glioblastomas (GBM) are treated with high doses of ionizing radiation (IR). A mechanism by which radiation therapy counterintuitively spurs recurrence in GBM was uncovered, and a novel senolytic strategy to prevent recurrence was tested.

- Senescent glioblastoma cells arising after radiotherapy secrete tumor-promoting factors that engender the senescence-associated secretory phenotype (SASP).
- SASP factors secreted by senescent cells activate the JAK-STAT3 and NF- κ B pathways in non-senescent tumor cells, promoting proliferation, therapy resistance and SASP spreading.
- Senescent tumor cells rely on the anti-apoptotic protein cIAP2 for survival and can, therefore, be eliminated by targeting cIAP2

with birinapant.

• Birinapant, given after radiotherapy, can delay or prevent glioblastoma recurrence in mouse tumor models by reducing therapy-induced senescence (TIS)."

-Thank you for providing a synopsis image, however we need a graphical abstract, including more information on the study and results. You may refer to any of our published articles for reference. Please upload it as a 550 px wide x 300-600 px high file. A cropped portion of this image will serve as thumbnail for the table of content on our webpage.

7/ As part of the EMBO Publications transparent editorial process initiative (see our Editorial at <http://embomolmed.embopress.org/content/2/9/329>), EMBO Molecular Medicine will publish online a Review Process File (RPF) to accompany accepted manuscripts.

This file will be published in conjunction with your paper and will include the anonymous referee reports, your point-by-point response and all pertinent correspondence relating to the manuscript. Let us know whether you agree with the publication of the RPF.

I look forward to receiving your revised manuscript.

With kind regards,

Lise Roth

***** Reviewer's comments *****

Referee #1 (Comments on Novelty/Model System for Author):

The authors adequately addressed my previous critique. The manuscript is improved.

Referee #1 (Remarks for Author):

The authors adequately addressed my previous critique. The manuscript is improved. One minor issue that needs to be fixed is the labelings in Figure S1,c,d,e. The prefix "si" that refers to the siRNAs in the legend (e.g. siCIAP1) is nowhere to be found in the figure X axis labeling, where the siRNAs appear to be referred to ciAP2-1 or ciAP2-2. This is confusing and should be changed.

Referee #2 (Comments on Novelty/Model System for Author):

This study is well performed and the results are relevant for the scientific community

Referee #2 (Remarks for Author):

The authors answered all my comments and suggestions

Response to editorial suggestions

Jan 24, 2025

Dear Dr. Roth,

We are delighted to learn that our revised manuscript is acceptable for publication in EMBO Molecular Medicine, pending revisions to address editorial issues. We have made the requested changes, as detailed in the point-by-point response below.

We are very grateful to you for your advice and support throughout the submission and revision process.

Best regards,

Sandeep

***** Editor's comments *****

1/ Referees' comments:

Please address the remaining comments from referee #1.

“One minor issue that needs to be fixed is the labelings in Figure S1,c,d,e.” The reviewer suggested changes to the x-axis labels of these figures so that it is clear that siRNAs are being used to target cIAP2. We have made these changes to increase clarity.

2/ Manuscript text:

- Please indicate in track changes mode any new modification. **A clean copy and a copy with track changes has been uploaded.**

- Authors: there is a discrepancy between Jair Machado Espindola Netto in the manuscript text and Jair Machado Espindola in the submission system, please correct. **“Jair Machado Espindola Netto” in the manuscript is correct.**

- We can accommodate a maximum of 5 keywords, please adjust accordingly. **Adjusted.**

- Methods:

o Thank you for providing a Reagents and tools table. Please remove it from the manuscript file and upload it as a separate file. **Removed and uploaded as a “Reagents and tools table” file**

o Cell culture: please indicate whether the cells were authenticated. **Cell lines and PDX cultures were directly purchased from ATCC and the Mayo Clinic and were not authenticated. This is now indicated in the “Cell culture” section.**

o Mice: please provide housing and husbandry conditions **These details have been added.**

o Statistical analysis: please provide a statement on exclusion/inclusion criteria, blinding and randomization. **These details have been added.**

- Data Availability: Thank you for depositing your datasets. Please note that they must be made public before acceptance of the manuscript. **The datasets have been made public.**

- Acknowledgements: Please note that the information provided in the manuscript and the submission system should match, please adjust accordingly. **The information has been adjusted.**

- Please add a heading: "Disclosure statement and competing interests". Please review our updated policy <https://www.embopress.org/competing-interests> and update your competing interests if necessary. **Added.**

3/ Figures and Appendix:

- During our standard figure check, we noticed anomalies in Figure EV5C. Please carefully check this figure panel composition, clarify and update if needed. **Thank you for catching this error. Apparently, a wrong panel was unintentionally inserted for GBM123/IR/Biri/Day0 resulting in an inadvertent image duplication in the GBM123 set. We apologize for the error. We would like to point out that we had uploaded the correct source data for these panels; the error occurred during figure assembly and was limited to this single panel. This has now been corrected.**

- Please rename the Appendix tables "Appendix Table S1" etc. and update the callouts in the manuscript text accordingly. Please upload the final version as a PDF document. **Done**

- Please address the queries from our copy editors in the figure legends:

1. Please note that the exact p values are not provided in the legends of figures 1A, D, E; 2C; 3D, F, I; 5A, C; 6C; EV1 A, B, D; EV2 B; EV3 B, D, G; EV5 B **We have now added exact p values to the legend. For plots that have more than 10 p values, we have generated an Appendix Table S4 with these values and have referred to the table in the figure legend.**

2. Please indicate the statistical test used for data analysis in the legends of figures 1A, C, D, E; 2C; 3D, F, I; 5A, C; 6C; EV1 A, B, D; EV2 B; EV3 B, D, G; EV5 B. **This is now indicated.**

3. Although 'n' is provided, please describe the nature of entity for 'n' in the legends of figures 1A, C, D, E; 2C, 3B, D, F, I; 4C; 5A, C; 6C, E; EV1 A, B, D; EV2 B; EV3 B, D, G, I; EV4 C, EV5 B **This is now described.**

4. Please note that the measure of center for the error bars needs to be defined in the legends of figures 1A, D, E; 2C, 3B, D, F, I; 4C; 5A, C; 6C, E; EV1 A, B, D; EV2 B, EV3 B, D, G, I; EV4 C, EV5 B. **This is now noted in the legends.**

4/ Source Data: Thank you for providing detailed Source Data. We would appreciate if you could submit the numeric files as excel documents, not prism files. **Source data have been submitted as excel files.**

5/ Checklist:

- Experimental animals, housing and husbandry conditions: please fill in. **Done**

- Experimental study design and statistics: please fill in the sections on blinding and inclusion/exclusion criteria. **Done**

6/ Synopsis:

- I introduced minor edits in your synopsis, please let me know if you agree or amend as you see fit:

"Glioblastomas (GBM) are treated with high doses of ionizing radiation (IR). A mechanism by which radiation therapy counterintuitively spurs recurrence in GBM was uncovered, and a novel senolytic strategy to prevent recurrence was tested.

- Senescent glioblastoma cells arising after radiotherapy secrete tumor-promoting factors that engender the senescence-associated secretory phenotype (SASP).

- SASP factors secreted by senescent cells activate the JAK-STAT3 and NF- κ B pathways in non-senescent tumor cells, promoting proliferation, therapy resistance and SASP spreading.

- Senescent tumor cells rely on the anti-apoptotic protein cIAP2 for survival and can, therefore, be eliminated by targeting cIAP2 with birinapant.

- Birinapant, given after radiotherapy, can delay or prevent glioblastoma recurrence in mouse

tumor models by reducing therapy-induced senescence (TIS)." **Thank you for the edits**

-Thank you for providing a synopsis image, however we need a graphical abstract, including more information on the study and results. You may refer to any of our published articles for reference. Please upload it as a 550 px wide x 300-600 px high file. A cropped portion of this image will serve as thumbnail for the table of content on our webpage. **A graphical abstract has been uploaded.**

7/ As part of the EMBO Publications transparent editorial process initiative (see our Editorial at <http://embomolmed.embopress.org/content/2/9/329>), EMBO Molecular Medicine will publish online a Review Process File (RPF) to accompany accepted manuscripts. This file will be published in conjunction with your paper and will include the anonymous referee reports, your point-by-point response and all pertinent correspondence relating to the manuscript. Let us know whether you agree with the publication of the RPF. Please note that the Authors checklist will be published at the end of the RPF. **We agree.**

***** Reviewer's comments *****

Referee #1 (Comments on Novelty/Model System for Author):

The authors adequately addressed my previous critique. The manuscript is improved.

Referee #1 (Remarks for Author):

The authors adequately addressed my previous critique. The manuscript is improved. One minor issue that needs to be fixed is the labelings in Figure S1,c,d,e. The prefix "si" that refers to the siRNAs in the legend (e.g. siCLAP1) is nowhere to be found in the figure X axis labeling, where the siRNAs appear to be referred to clAP2-1 or clAP2-2. This is confusing and should be changed.

Referee #2 (Comments on Novelty/Model System for Author):

This study is well performed and the results are relevant for the scientific community

Referee #2 (Remarks for Author):

The authors answered all my comments and suggestions

5th Feb 2025

Dear Prof. Burma,

Thank you for submitting your revised files. I am pleased to inform you that your manuscript is accepted for publication and is now being sent to our publisher to be included in the next available issue of EMBO Molecular Medicine!

Yours sincerely,
